# Sediment discharge from Greenland's marine-terminating glaciers is linked with surface melt

Camilla S. Andresen ®[1] ✉, Nanna B. Karlsson ®[1], Fiammetta Straneo ®[2], Sabine Schmidt ®[3], Thorbjørn J. Andersen ®[4], Emily F. Eidam[5], Anders A. Bjørk ®[6], Nicolas Dartiguemalle ®[1], Laurence M. Dyke[1], Flor Vermassen ®[6] & Ida E. Gundel[1]

Sediment discharged from the Greenland Ice Sheet delivers nutrients to marine ecosystems around Greenland and shapes seafloor habitats. Current estimates of the total sediment flux are constrained by observations from land-terminating glaciers only. Addressing this gap, our study presents a budget derived from observations at 30 marine-margin locations. Analyzing sediment cores from nine glaciated fjords, we assess spatial deposition since 1950. A significant correlation is established between mass accumulation rates, normalized by surface runoff, and distance down-fjord. This enables calculating annual sediment flux at any fjord point based on nearby marine-terminating outlet glacier melt data. Findings reveal a total annual sediment flux of 1.324 +/− 0.79 Gt yr-1 over the period 2010-2020 from all marine-terminating glaciers to the fjords. These estimates are valuable for studies aiming to understand the basal ice sheet conditions and for studies predicting ecosystem changes in Greenland's fjords and offshore areas as the ice sheet melts and sediment discharge increase.

The sediment transported by Greenland's marine-terminating glaciers and subsequently deposited in the ocean is important for both scientific research and societal purposes. For years, geoscientists have used sediment archives to reconstruct past changes in the extent and dynamics of ice sheets, which provide a critical context for ongoing instrumentally observed ice sheet changes and contribute to a better understanding of the underlying processes[1].

In recent years, there has been a growing emphasis on accurately quantifying the sediment flux from the ice sheet, highlighting the need for precise measurements[2]. For instance, the impact of sediment accumulation near marine-terminating glacier margins can have significant implications for glacier dynamics and stability, as the formation of shoals (moraines) from this sediment can buttress glaciers and/

or shelter them from incoming warm Atlantic Waters[3–7]. Thus, depending on the quantity of sediment deposited, this may have repercussions for future sea-level rise driven by ice sheet loss.

Moreover, the sediment carries nutrients and trace elements that may sustain marine ecosystems further offshore[8–13], although understanding the specific role of ice sheets and glaciers in marine nutrient cycling is still a field in its early days[14]. It has been suggested that marine-terminating glaciers play a crucial role in regulating nutrient availability for primary production in the photic zone of Greenland fjords through plume-induced upwelling of remotely sourced nitrate and dissolved phosphate[14] and subglacial discharge that contains bioessential elements such as $Si^{13,15,16}$ and $Fe^{10,17,18}$. Si stimulates the growth of diatoms, which are a primary food source for secondary

[1]Geological Survey of Denmark and Greenland, Department of Glaciology and Climate, Øster Voldgade 10, 1350 Copenhagen K, Denmark. [2]Scripps Institution of Oceanography, UCSD, La Jolla, USA. [3]CNRS, Univ. Bordeaux, Bordeaux INP, UMR 5805, F-33600 Pessac, France. [4]Department of Geosciences and Natural Resource Management, Univ. of Copenhagen, 1350 Copenhagen K, Denmark. [5]Oregon State University, Burt Hall 218, 2651 SW Orchard Avenue, Corvallis, OR 97331, USA. [6]Department of Geological Sciences, Stockholm University, 106 91 Stockholm, Sweden. ✉e-mail: csa@geus.dk

producers that support higher trophic levels in the ecosystems. In the subpolar North Atlantic, Si[19] and occasionally Fe[20] are the main limiting nutrients for diatom growth, with their concentrations in the water regulated by subpolar gyre dynamics and the amount of Si imported from Arctic rivers[21]. However, recent research has highlighted the potential for significant export of Si from the Greenland Ice Sheet, showing that glacial meltwater can contain up to 0.20 Tmol year⁻¹ of dissolved and dissolvable Si, which corresponds to ~50% of the dissolved Si transported by Arctic rivers[15]. This is consistent with evidence that the rate of Si dissolution in glacial environments may be higher than previously believed[22,23]. Similarly, it has been found that the ice sheet contributes ~15% of the total reactive phosphorus input to the Arctic oceans[24]. Understanding, and quantifying the release of sediment from the Greenland Ice Sheet into the oceans is, therefore, critical for predicting future changes to North Atlantic and Greenland fisheries.

Sediment from the Greenland Ice Sheet is delivered to the ocean via proglacial rivers[25], subglacial discharge at depth, and submarine melting of icebergs and glacial margins[3,26,27]. Fjord circulation, as well as iceberg drift, can distribute this sediment load far from the glacier front. Multiple studies have investigated the controlling mechanism for sediment production and/or transport from glaciers to fjords, e.g., [2,28]. However, there are no long-term datasets monitoring sediment transport from the marine margins of the Greenland Ice Sheet, either by iceberg-rafting, or from subglacial plumes, which emanates deep beneath the marine sectors of the ice sheet. Installing instrumentation to monitor sedimentation in these harsh environments is both technically challenging and dangerous[29]. Additionally, subglacial meltwater plumes, emanating from the grounding line of major outlet glaciers, vary in both time and space[30,31], making it difficult to robustly quantify sediment production from the ice sheet by monitoring alone.

Due to the lack of sediment flux data from the marine ice margins, previous ice sheet-scale constraints of sediment flux to the ocean have focused on terrestrial proglacial rivers in southwest and northeast Greenland and linked a single comprehensive data set of suspended sediment load within a proglacial river to satellite images of suspended sediment plumes[2]. The authors concluded that the Greenland Ice Sheet is a vast contributor of sediment to the ocean and provides 8% of the global ocean sediment budget despite supplying only 1.1% of the freshwater. However, the sediment flux from marine-terminating glaciers was measured indirectly using satellite imagery of surface waters. Given that approximately three-quarters of the mass loss from the Greenland Ice Sheet occurs through marine-terminating outlet glaciers[32], and that much of the discharge occurs deep in fjords near the grounding line, it is essential to investigate the connection between ice sheet melt and marine-terminating glaciers' sediment flux using additional approaches such as seafloor sampling. This is crucial for accurately quantifying the amount of sediment generated by the ice sheet and understanding the spatial distribution of sediment dispersion in the fjords surrounding Greenland.

To address the lack of observationally-constrained sediment flux to fjords with marine-terminating glaciers, we present data from 30 marine sediment cores collected from nine fjords with marine-terminating glaciers around Greenland. We use the mass accumulation rate (MAR) to infer the average glacial sediment flux and spatial pattern of sedimentation in these Greenland fjords since 1950. We apply a regression analysis and show that there is an empirical exponential relationship between surface meltwater runoff from marine-terminating outlet glaciers and the subglacial sediment flux to a given distance down the fjord.

## Results and discussion
### Subglacial sediment flux from a major Greenland outlet glacier
First, we use a marine sediment core dataset from the Sermilik Fjord by Helheim glacier in SE Greenland to document the spatial pattern of

sedimentation within a typical glaciated fjord in Greenland. Like hundreds of marine-terminating glaciers around Greenland, Helheim Glacier is connected to the continental shelf by a long, deep glacial fjord (Fig. 1b) into which the glacier delivers icebergs and meltwater[29,33]. Glacial meltwater, derived from surface and submarine melt, enters Sermilik Fjord hundreds of meters below sea level at the glacier grounding line[34,35], and drives buoyant upwelling plumes that entrain ambient waters and reach neutral buoyancy in the stratified upper water column[35–38]. Qualitative measurements of suspended sediment indicate that these plumes are highly effective at injecting and transporting sediment within the fjord[35] consistent with findings from other marine-terminating glaciers in Greenland[32,39,40] and modeling studies[41].

Specifically, we examined the sediment facies in thirteen sediment cores from Sermilik Fjord to assess both the spatial distribution of sediment types and the amount of sediment accumulated during the period 1950–2009 (Fig. 1 and Supplementary Figs. 2, 3). Sediment mass accumulation rates (MAR, kilogram dry sediment m⁻² seabed year⁻¹) were calculated as the product from ²¹⁰Pb-based sediment accumulation rates and dry bulk density (Methods, Supplementary information, source data file[42]).

In Sermilik Fjord, the sediment deposited near the glacier comprise fine-grained laminated clays and silts that are accumulated at relatively high rates ranging broadly between 20 and 100 kg sediment m⁻² yr⁻¹ (Fig. 1a). This sedimentary facies, referred to as plumites, is typically formed through suspension settling from glacial meltwater plumes[3,43] Down-fjord from the calving front of Helheim Glacier, mass accumulation rates decline exponentially to values of 2–4 kg sediment m⁻² yr⁻¹, indicating a diminishing influence of meltwater plume sedimentation in this area. Here, the sedimentary facies consists of mud with a high content of randomly located sand particles in a structureless matrix, called diamicton (Fig. 1a and Supplementary Figs. 2, 3). The sand grains are too heavy to remain suspended in the meltwater plume and were therefore transported to this part of the fjord, together with clay and silt, in icebergs[44]. It is worth noting that the quantity of clay and silt transported by icebergs to this location is expected to be smaller than that carried by the plume. This expectation is grounded in reports of 50–80% sand in basal ice debris by Russel Glacier[45] and 50–70% sand in icebergs sampled off Scoresbysund glaciers[46]. In comparison, the sand content in the diamicton at core sites ER07 and ER11 is only 20%[44] and therefore suggests clay and silt input from the plume. However, future sediment sampling from icebergs[47] is needed to robustly quantify the grain size distribution in iceberg sediments.

While there is no significant relationship between the accumulation rate of sand and core site distance to the glacier margin (Fig. 1b), higher sand flux rates are expected in the region underneath the 10–20 km long ice mélange (from which we do not have sediment cores), because icebergs tend to reside for a longer time here[48]. But in summary, substantially lower accumulation of sand, relative to plume sediment (Fig. 1a), supports previous suggestions that icebergs only supply a minor fraction of sediment to the ocean[2], although they may be more efficient at transporting sediment offshore in the Greenland vicinity. In contrast, the high rate of meltwater plume sediment deposition in the upper reaches of the fjord system suggests that the processes that are responsible for plume formation also control sediment flux from marine-terminating outlet glaciers. Our data indicate that the primary sediment transporter to the fjord is the subglacial discharge at the glacier's grounding zone, which is then routed into the fjord via buoyant meltwater plumes and through fjord circulation, transporting the sediment over considerable distances downstream.

### Sediment flux is linked to surface runoff
We extend our dataset and calculate the MAR in cores from a variety of other Greenlandic fjords with marine-terminating glaciers to test whether the exponential relationship established between distance

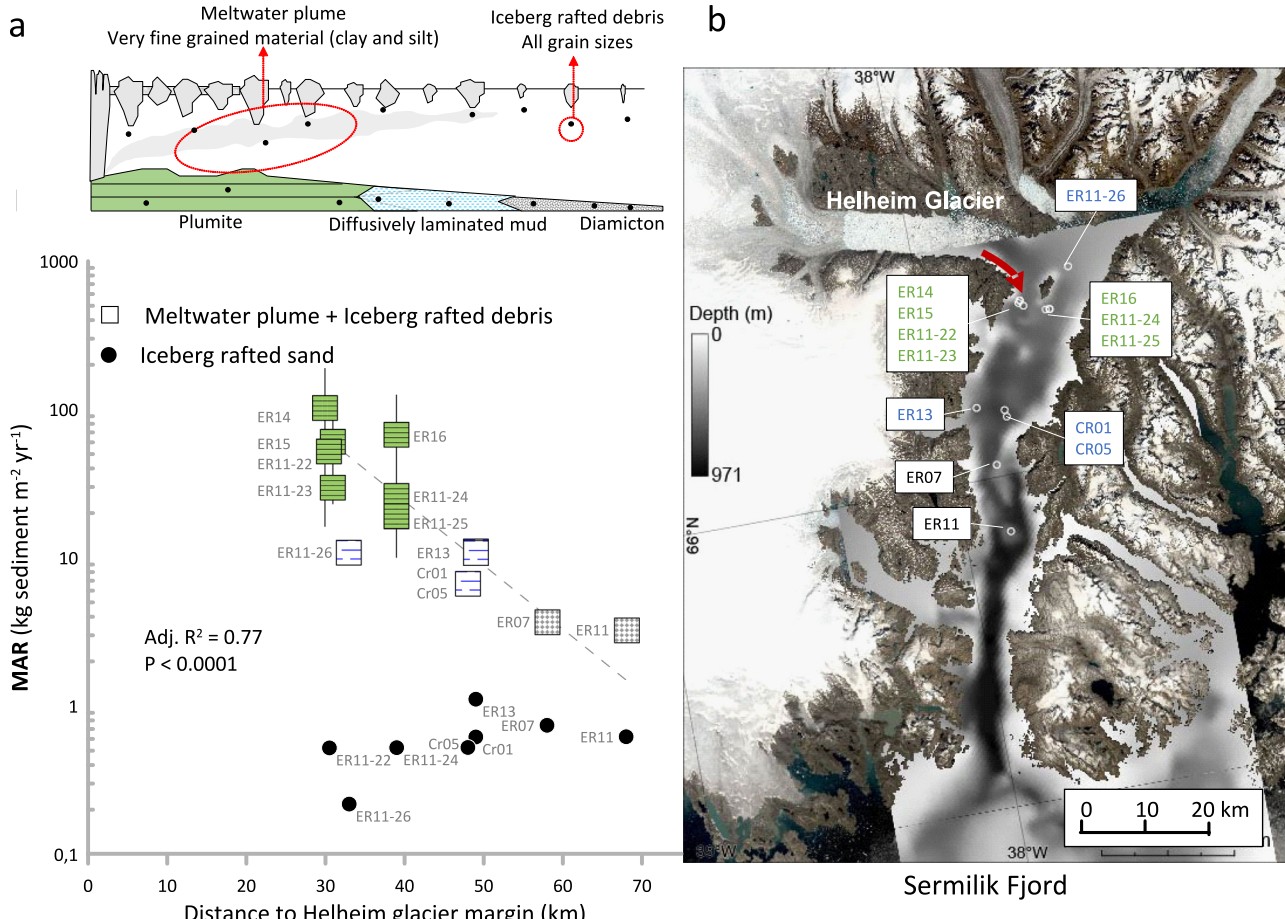

**Fig. 1 | Mass accumulation rate in Sermilik fjord decreases with distance from the Helheim Glacier margin. a** The conceptual model shows the influence of the subglacial plume and iceberg melting on the sedimentary facies and the mass accumulation rate (MAR) magnitude. Below is the late 20th century (1950–2009) average MAR of Sermilik Fjord sediments. Squares: MAR of bulk material in the size fraction <1000 μm with color legend implying the sedimentary facies (Supplementary Fig. 2). Stippled line: Exponential fit adjusted $R^2 = 0.77$ and $P$ value <0.0001 ($n = 13$). Solid circles: MAR of iceberg rafted sand in the size fraction 63–1000 μm. The distance is measured to the glacier margin position in the year of core collection (source data file[42]). The error bars of the data points are quantified to 10% but are further detailed in the method section. **b** Sermilik Fjord with sediment core locations. The red line indicates the main track of icebergs and meltwater flowing out of the fjord[48]. Note that core ER11-26 is positioned off this track, which explains the low MAR values. Background imagery is a mosaic of pan-sharpened Landsat 8 scenes[85]. Bathymetric data of Sermilik Fjord are from a compilation of single-beam echo-sounding surveys[86]. Data are provided as a Source Data file[42].

from the glacier margin and sediment accumulation rate in Sermilik Fjord is representative of glaciated fjords in Greenland in general (Fig. 2 and Methods). We compile the average MAR for the period 1950–2009 from eight other fjords into which the following major glaciers terminate: Kangerdlussuaq Christian IV, Thrym, Eqalorutsit Kangilliit Sermiat, Kangiata Nunaata Sermia (KNS), Sermeq Kujalleq, Kangilerngata, and Upernavik, (Supplementary Fig. 4). All sediment cores are located within 80 km of the glacier margins. The analysis thus excludes sediment records from open coastal and continental shelf settings because hydrodynamic energy levels at the seabed are higher in these exposed areas and this will influence the clay and silt contribution to the MAR values either through winnowing or refocusing. Moreover, sediment cores show that the 20th-century sedimentation rates are generally orders of magnitudes lower on the continental shelf outside of the investigated fjord settings[49–51]. In fjords with multiple marine-terminating outlet glaciers, we measured the core location distance from the glacier with the highest meltwater and iceberg production (Supplementary Fig. 4 and source data file[42]). Some glaciers around Greenland advanced substantially during the Little Ice Age, and subsequently retreated in the early 20th Century (Supplementary Fig. 4 and source data file[42]); thus, rates were evaluated post-1950.

The sediment data from fjords around Greenland confirms the pattern of an exponential decrease in MAR with distance from the glacier front (Fig. 2b, statistically significant with $p$ value = 0.0001). However, the magnitude of the later 20th-century MAR at a given distance from the glacier varies between fjords. We hypothesize that this difference is primarily due to the substantial variation in subglacially discharged surface runoff from the different glaciers over the later 20th Century. In other words, we propose that sediments are abundant under marine-terminating glaciers, due to their high flow velocities (allowing substantial erosion) and large catchments (implying a large amount of available sediments). Thus, the limiting factor for sediment flux into the fjords is likely the transport mechanism that moves the sediments to the glacier front. Sediments are thought to be influenced by a variety of processes on their pathway from initial erosion of bedrock[2,28,52], through release from the subglacial system[53,54], and during their transport and accumulation within the fjord system. For example, the hydrological regime and porewater pressure underneath the ice in response to surface melt[55–57], influence ice dynamics and basal sliding[33], creating feedback to the ice velocity and erosion. Regardless of these complexities in subglacial sediment dynamics[58,59] the common driver/modulator is surface melt. Thus, it is reasonable to attribute MAR differences between fjords to differences

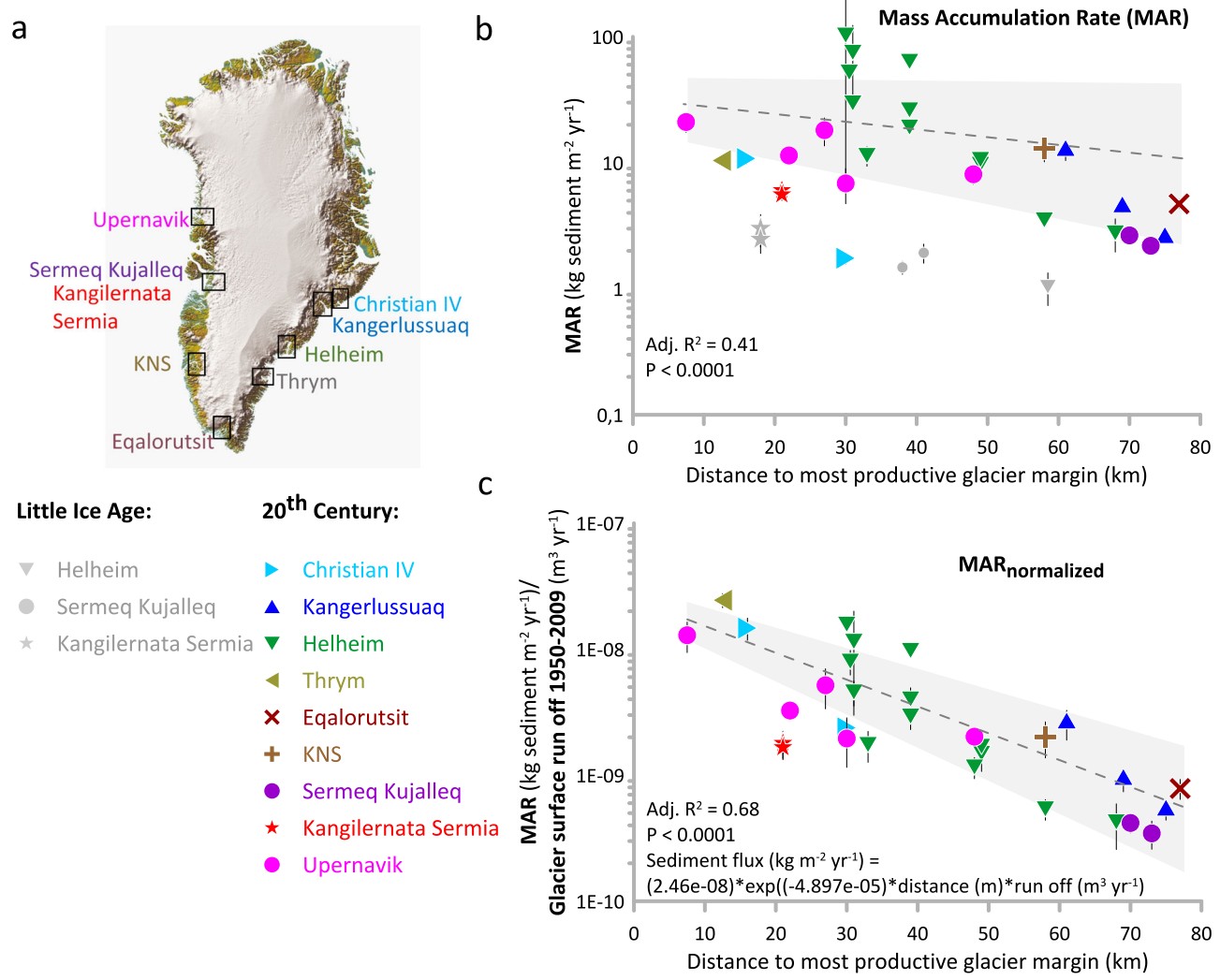

**Fig. 2 | Sediment flux from Greenland's marine-terminating glaciers, as inferred from the mass accumulation rate within fjords adjacent to these glaciers, exhibits a clear connection with surface melt processes. a** Map with glacier-fjord systems. **b** Average MAR (mass accumulation rate) in the later 20th century (1950–2009) as a function of distance from the glacier margin position in the year of coring (colored symbols) and average MAR during the Little Ice Age (1200–1900 CE) as a function of distance from the glacier margin position during the Little Ice Age (gray symbols) (Methods and Supplementary Fig. 3). Data error is quantified to 10% (Methods). Upper stippled line: Exponential fit to later 20th-century data points (gray shading 95% confidence level) with adjusted $R^2 = 0.41$ and P value <0.001 ($n = 30$). **c** The later 20th century MAR is normalized according to average glacier surface runoff 1950–2009[64]. Exponential fit (gray shading 95% confidence level) with adjusted $R^2 = 0.68$ and P value <0.0001 ($n = 30$). Data error quantified to 20% (Methods). Map base from the Danish Geodata Agency. Data were provided as a Source Data file[42].

in surface runoff to the fjords. This link between sediment flux and surface melt is also in line with the fact that MAR values are an order of magnitude lower during the colder Little Ice Age (1200–1900 CE[60]), when subglacial runoff and calving were markedly lower than today[61–63] (Fig. 2b).

To account for differences in surface runoff between glaciers, the later 20th century (1950–2009) MAR values are thus normalized (Fig. 2c, Methods) by dividing MAR for individual core sites with the average surface runoff over the period 1950–2009[64] from all the glaciers (i.e. including side glaciers) that contribute sediment-loaded meltwater to the fjord (Source data file[42]). In the normalization, we assume that all surface runoff produced over the glaciers' catchments[65] enters the subglacial system and is discharged subglacially at the glacier front, which is a common assumption in most studies aiming to quantify meltwater outflux[f.ex64]. The assumption is because most of the englacial transport of water in Greenland glaciers takes place in moulins (vertical pipes connecting the surface of the ice

with the bed) or in crevasses[66]. Studies have shown that surface meltwater penetrates to the bed of the ice comparatively fast (hours to days[67]).

The normalization procedure yields a statistically highly significant correlation (adjusted $R^2 = 0.68$ with $p$ value <0.001), thus supporting our hypothesized exponential relationship between MAR at a certain location in the fjord, relative to both the core location's distance to the margin of the most productive glacier and to the surface runoff from the glacier catchment:

$$MAR_{normalized} = a * exp(b*D) \qquad (1)$$

Where $D$ is distance to the glacier margin in meters, and a and b are the best-fit parameters (a = 2.460e-08 and b = −4.897e-05).

Thus, the equation allows calculation of the annual sediment flux (MAR) at any specific location along the axis of a glacial fjord using just the surface runoff from the local marine-terminating outlet glaciers

contributing melt water to the fjord:

$$MAR = (2.460e-08)*\exp((-4.897e-05)*D)*R \quad (2)$$

MAR = Annual sediment flux (kg sediment deposited pr m$^2$ seabed yr$^{-1}$)

$D$ = Distance to glacier margin (m)

$R$ = Annual surface runoff (m$^3$ yr$^{-1}$).

The confidence interval is large near the glacier margin due to a lack of data points but decreases 10–15 km from the margin and farther down-fjord (note the logarithmic y-axis scale).

The empirical relationship outlined in Eq. (2) implies that sediment flux at a certain location in a glacial fjord is modulated primarily by glacier melt processes. Other processes may influence MAR at a certain location in the fjord. These processes may include local sediment focusing/dispersal in response to fjord narrowing/widening, respectively, or the strength of the hydrographic regime at the seabed. Implicitly, we are assuming that most of the sediment flux, which is associated with the subglacial discharge, occurs during summer when the plume is more likely to dominate the fjords' hydrographic circulation[68,69]; however, other drivers of circulation in fjords may also influence currents and transport of sediments[69]. For example, the relatively high MAR of sediment cores at the bend in Sermilik fjord (Fig. 1a core ER14), may indicate a decrease in the current strength, as a result of complex bathymetry, that may also play a role in the deposition of sediments here.

Through the examination of sedimentary facies, our analysis was specifically directed toward assessing the accumulation of sediment originating from the meltwater plume and iceberg rafting (Methods). Thus, the sediment we have tracked has not undergone redeposition due to mass wasting events or turbidite flows, which are important processes for near-bed transfer of glacial sediment into the marine environment[70]. It's important to underscore that these are secondary processes, and by omitting sediment with turbidites, we can focus on assessing the primary accumulation of plume and iceberg sediment.

Despite these considerations, we emphasize that the combined data set provided here demonstrates that down-fjord sediment flux varies with distance and subglacial discharge in a statistically significant empirical relationship. This robust first-order quantification is

valuable in view of the lack of long-term instrumented measurements of sediment production from Greenland's marine-terminating glaciers. Moreover, we highlight that our approach provides the integrated signal from all plumes within a fjord and over many years and propose this explains the robust relationship.

In addition to providing an observationally constrained estimate of sediment flux from marine-terminating glaciers, we also demonstrate that a significant portion of the sediment flux from the ice sheet is confined to a 80-100 km zone down-fjord, coinciding with an area of increased nutrient upwelling which has been linked to high primary productivity in glacial fjords around Greenland[16,71–73]. Future ocean data sampling campaigns in Greenland fjords should ideally incorporate sediment coring for the assessment of additional MARs (especially from sites closer to the glaciers), and thereby enable the scientific community to continually enhance the reliability of these data.

## Erosion rate and sediment flux from Greenland's marine-terminating glaciers

We use the relationship described by Eq. (1) to calculate the total annual sediment flux from the nine glaciers in our study for the period 1950–2009. We do this by calculating the annual amount of sediment deposited per square meter of seabed along the fjord, and (assuming that this deposition rate is representative across the fjord), we estimate the total mass of sediment deposited (see Eq. (4), Methods). This gives a total sediment flux from the nine glaciers of 0.210 ± 0.137 Gt yr$^{-1}$ (Table 1).

We then utilize our estimated sediment yield to calculate the erosion rates of each glacier catchment (Table 1, Methods, source data file[42]), giving estimates ranging from 0.04 mm yr$^{-1}$ (Upernavik catchments) to 0.4 mm yr$^{-1}$ (Sermeq Kujalleq catchment). The estimates represent catchment-wide averages and are likely to mask substantial spatial variability[cf74]. Studies of the Greenland Ice Sheet have reported erosion rates of 1.0 ± 0.5 mm yr$^{-1}$ (average ice sheet erosion rate[74]), 0.08 and 0.17 mm yr$^{-1}$ (southwest and west Greenland land-terminating glaciers, respectively[74,75]), 0.26 ± 0.16 mm yr$^{-1}$ (Sermeq Kujalleq[76]) and 0.29–0.34 mm yr$^{-1}$ in northwestern Greenland[77]. Thus, our results, based on sediment cores, are in good agreement with previous studies.

## Table 1 | Sediment flux from Greenland's marine-terminating glaciers and erosion rates

| Glacier[a] | Fjord | Sediment Flux (Gt yr-1)[b] | ±[c] | Erosion rate (mm yr-1)[f] | ±[g] |
|---|---|---|---|---|---|
| Upernavik | Upernavik | 0.0051 | 0.0033 | 0.04 | 0.026 |
| Thrym | Qimutuluittiip Kangertiva (Skjoldungen) | 0.0008 | 0.0005 | 0.17 | 0.106 |
| Christian IV | Nansen Fjord | 0.0035 | 0.0023 | 0.10 | 0.066 |
| Kangilerngata | Ikerasak (Ata Sund) | 0.0057 | 0.0037 | 0.26 | 0.169 |
| Helheim | Sermilik | 0.0425 | 0.0277 | 0.26 | 0.169 |
| KNS | Kangersuneq | 0.0153 | 0.01 | 0.20 | 0.130 |
| Kangerlussuaq | Kangerlussuaq | 0.0423 | 0.0276 | 0.27 | 0.176 |
| Sermeq Kujalleq | Qeqertarsuup Tunua (Disko Bugt) | 0.0841 | 0.0549 | 0.39 | 0.255 |
| Eqalorutsit | Ikersuaq (Brede Fjord) | 0.0109 | 0.0071 | 0.38 | 0.248 |
| Combined | | 0.2103 | 0.1374 | | |
| Upscaled to all marine terminating glaciers (1950–2009)[d] | | 0.911 | 0.544 | | |
| Upscaled to all marine terminating glaciers (2010–2020)[e] | | **1.324** | **0.791** | | |
| Total Greenland ice sheet (1999–2013)[2] | | 1.154 | 0.636 | | |

[a]The most melt water productive of the glaciers contributing sediment to the fjord.
[b]Sediment flux from all glaciers contributing melt water to the fjord. Each glacier contribution derived by double integrating sediment flux across all sediment cores (Methods).
[c]Uncertainties derive from the covariance of the fit parameters from the exponential function and from the 15% uncertainty in the surface meltwater estimates (Methods).
[d]The nine glacier sites contribute 25% of total surface runoff from Greenlands' marine-terminating glaciers (Methods).
[e]GrIS surface melt increased by 45% from 2010–2020 compared to 1950–2009[64].
[f]Erosion rate derived as sediment flux from all glaciers contributing to the fjord divided by the catchment area of the glaciers[28] (Methods).
[g]Uncertainty derived from the uncertainty of the sediment flux[c] (Methods).

We consider that Eq. (1) captures the governing processes for sediment deposition from marine-terminating glaciers in general, and thus the proposed relationship between sediment deposition, distance to the glacier front, and surface runoff should apply to other marine-terminating glaciers that share similarities with the nine glaciers in this study. We hypothesize that this is the case for all marine-terminating glaciers in Greenland, as they are typically characterized by high ice flow velocities and large catchments, and thus the available surface runoff will dictate the sediment flux from the glaciers. We can therefore apply Eq. (1) to upscale our estimate for the entire section of the Greenland Ice Sheet that discharges through marine-terminating outlets. This yields an estimate of $0.911 \pm 0.544$ Gt yr$^{-1}$ (1950–2009) (Table 1, Methods). This estimate seemingly aligns well with a previous estimate of $1.154 \pm 0.636$ Gt yr$^{-1}$ (1999–2013) of sediment from the whole ice sheet[2]. However, given that our results only include sediments from marine-terminating glacier, this suggest that the previous study may have underestimated the sediment output from the marine-terminating sector of the ice sheet[2]. In fact, the previous study, which relied on satellite observations of sediment plumes, may not have fully accounted for sediment transport processes that occur deep in the fjord—e.g., sediment fallout within the water column before the plume reaches the surface where it can be detected by satellites[2]. This study also found only a weak linear correlation between sediment concentration and melt water discharge. It is worth considering that while sediment transport (by surface runoff) controls sediment flux from marine-terminating glaciers, sediment availability rather than sediment transport may be the limiting factor for land-terminating glaciers. On average, the catchments of marine-terminating glaciers are 20% larger than those of land-terminating glaciers, and average velocities are twice as high (Methods). This implies an overall greater sediment production by erosion and a larger catchment area for marine-terminating glaciers. We acknowledge that the erosion rate values presented here (Table 1) and previously reported[74–77] represent large local variability and do not exclude that individual land-terminating glaciers may have high erosion rates.

Our findings underscore the substantial influence of surface melt on the discharge of sediment from beneath the marine-terminating glaciers, indicating that an increase in surface runoff will result in a commensurate rise in sediment flux into the waters surrounding Greenland. In recent decades (2010-2020), the surface runoff has significantly increased with a decadal average of 230 km$^3$ (ref. 64). Previous studies have suggested an exponential relationship between erosion rates and precipitation, implying that an increase in surface runoff will lead to a substantial increase in sediment flux[78]. Here we take a conservative approach and assume that the sediment flux increases linearly with runoff volumes. This implies an increase in sediment and nutrient transport from the ice sheet into the oceans in recent years, with an estimated increase of up to 45% and a combined total of $1.324 \pm 0.79$ Gt yr$^{-1}$ currently being overturned from the marine-terminating sector (Table 1). This is consistent with observations indicating that the present-day sediment flux from Greenland is approximately 56% higher than during 1961–1990[2].

Climate models unanimously predict an increase in annual surface melt[79], and the effects of this increase may be far-ranging. Increased primary productivity from increased Si[80] and Fe[17,81], combined with plume-induced nitrate and phosphate upwelling[e,q13,16,18]. may cascade to the highest trophic level, including the habitat used by top predators, such as seabirds and seals[82]. However, increased sediment loads in the water could also reduce light penetration and thereby impact bloom dynamics and timing towards reduced primary productivity[14]. Similarly, increased sediment flux from the Greenland Ice Sheet warrants an improved understanding of the factors that control the bioaccessibility of Si and Fe, including in dissolved form, as well as their transport to and within the North Atlantic Ocean, in terms of conserving marine resources far off Greenland.

Unraveling the precise consequences for the biological system, and hence for the societies that rely on them, will require further studies. Here we provide a first-order quantification of a significant modulator of the biogeochemical state of the ocean waters; the sediment flux from Greenland's marine terminating glaciers to the ocean. Our results can serve as a first-order validation of numerical ice sheet and fjord circulation models, that simulate sediment production, retention, and dispersal within glaciated fjords.

## Methods

### Mass accumulation rate (MAR) calculations using sediment cores

We have calculated mass accumulation rates for the later 20th century (1950–2009) from 27 marine sediment cores from fjords by marine terminating calving glaciers and have supplemented the data with previously reported mass accumulation rates from three additional cores (Supplementary Fig. 4 and source data file[42]). The selection criterion for our analysis is that the cores are taken within 100 km from the current margin. We have applied the Constant Flux-Constant Sedimentation (CF:CS) model to the $^{210}$Pb profiles from the 27 cores to derive the average sedimentation rate (SAR) for the period 1950–2009, since most cores were retrieved from 2009–2014. The sediment water content was measured concurrently with sampling for $^{210}$Pb analysis, allowing us to calculate the average mass accumulation rates (MAR) from the 27 cores in the period 1950–2009 (source data file[42]):

$$\text{Average MAR (kg m}^{-2}\text{yr}^{-1}\text{)} = \text{average SAR (m yr}^{-1}\text{)}^* \text{dry bulk (kg m}^{-3}\text{)}$$

(3)

Dry bulk density is calculated as ((100-average % water)/100) * 2650 kg m$^{-3}$, where 2650 kg m$^{-3}$ is the density of quartz.

The $^{210}$Pb profiles that provide input to the SAR and MAR calculations includes previously reported measurements from 19 cores and new measurements from eight cores (Supplementary Fig. 1 and source data file[42]).

The sedimentary facies analysis for cores from Sermilik Fjord by Helheim Glacier (Fig. 2a) was conducted using x-ray imaging of the sediment cores and, in some cases, grain size analysis on subsamples (Supplementary Fig. 2).

Some of the cores are also dated in the interval older than the 20th century using the $^{14}$C dating method. This allowed us to compare the average MAR of the later 20th century with the average MAR during the cold Little Ice Age (1200–1900 CE) (overview in source data file[42]).

The data provide the average MAR in the 1950–2009 period. Short term temporal changes in MAR within this period cannot be accurately determined since the CF:CS method provides a linear sedimentation rate.

**Error assessment MAR**. MAR error assessment (vertical error bar, Supplementary Fig. 5): The uncertainty in MAR results from the error in the calculated sedimentation rates (SAR). The error of SAR is obtained by propagating the error on the slope of the regression as SAR is calculated as the ratio of the decay constant to 'b,' which represents the slope of the exponential function. Except for cores ER14, ER15, and ER16 (Sermilik Fjord), which have marked MAR errors inherited from high sedimentation rate errors, the average uncertainty on the estimated MAR is, on average, ~10%. The average uncertainty on the Little Ice Age MAR is 7–15% and inherited from the error of the $^{14}$C dating (source data file[42]).

Error assessment distance to glacier front: (horizontal error bar, Supplementary Fig. 5): The relationship between the later 20th century average MAR and distance to most productive glacier builds on the assumption that the grounding line (GL) has been constant over the period 1950–2009. However, this is not the case for some of the glaciers. To assess the influence on the relationship between MAR and

distance to the glacier margin from changes to the grounding line, we plot the distance between the current GL and the 1950 GL as a horizontal error bar (in the direction towards the glacier margin). Applying a trend line fit using the distance between the core site and the 1950 GL, we find that the error introduced from grounding line changes does not significantly influence the reported relationship between 1950 and 2009 average MAR and distance to the most productive glacier terminus.

## Normalization of MAR values

We normalize the MAR values using surface runoff from the period 1950–2009[64] (source data file[42]).

**Error assessment normalized MAR.** The error on surface runoff is 10% (ref. [64]). Summarizing this with the uncertainty in the MAR values of 10% in the normalized MAR values, the uncertainty is quantified as ~20%.

## Exponential fits

We use the algorithm *curve_fit* from the open-source Python package SciPy to find the best-fit parameters for the datapoints. The adjusted $R^2$-values (uncentered) are calculated using the open-source Python module *statsmodels* to assess the goodness of fit.

## Calculation of the combined sediment flux from Greenland's marine margins

We utilize our empirical relationship to calculate the total flux of sediments discharged at our study sites from the marine-terminating glaciers Helheim, Kangerdlussuaq, Christian IV, Thrym, Eqalorutsit, KNS, Sermeq Kujalleq, Kangilerngata, and Upernavik Glaciers (gates in source data file[42]). We use all available sediment-core data, as exemplified by Eq. (1), arguing that this minimizes the effect of outliers, and we calculate the sediment flux from each glacier by a double integral of the relationship:

$$\int_{y0}^{y1} \int_{x=0km}^{x=\infty} s(x)\, dxdy = S \qquad (4)$$

Where s(x) is the relationship described in Eq. (4) and S is the total flux of sediment per meltwater volume for one glacier. The coordinates x,y describe along and across-fjord geometry, respectively, implying that y is equivalent to the glacier front width if the fjord does not narrow or widen. We then multiply with the average annual meltwater volume (1950–2009) for the individual glaciers to retrieve the sediment fluxes (Table 1). We obtain a sediment flux of 0.210 ± 0.137 Gt yr$^{-1}$.

To derive the sediment flux from all marine-terminating glaciers, we upscale the result from the nine glacier sites in our dataset. We build on the assumption that sediment availability is not a limiting factor for marine-terminating glaciers in Greenland and that sediment flux scales with surface runoff. To obtain a conservative estimate of total sediment flux, we upscale linearly by considering surface runoff volumes. The nine glacier sites contribute 25% (37 km$^3$) of total surface runoff from marine-terminating glaciers of the Greenland Ice Sheet (161 km$^3$)[64] providing a total estimate of 0.911 ± 0.544 Gt yr$^{-1}$ (1950–2009).

**Error assessment sediment flux.** The uncertainties derive firstly from the covariance of the fit parameters from the exponential function and secondly from the 15% uncertainty in the surface meltwater estimates.

## Estimation of erosion rates

We convert our sediment flux values (Table 1) from Gt yr$^{-1}$ to m$^3$ yr$^{-1}$ using a density of 2650 kg m$^{-3}$. The average basin-wide erosion rate is estimated by dividing the sediment yield by the area of the glacier catchment[28]. We use previously defined glacier catchments[83], and in the cases where multiple glaciers discharge into the same fjord, we sum the glacier catchment areas. See source data file[42] for the area of each glacier catchment.

## Glacier catchment sizes and velocities

We use previously published glacier catchments[83] to assess the glacier catchment size of marine-terminating and land-terminating glaciers. The catchments are defined using the direction of ice flow for fast-moving areas (>100 m/yr) and the steepest surface slope for slower areas, where the surface slope is smoothed over 10 ice thicknesses (see ref. [83]). We only consider glaciers that are part of the inland ice disregarding peripheral ice caps and glaciers. The average catchment size is 6970 km$^2$ for marine-terminating glaciers and 5830 km$^2$ for land-terminating glaciers. Moreover, 36 marine-terminating glaciers have a surface area exceeding 10,000 km$^2$ (totaling 1,074,457 km$^2$) compared to only eight land-terminating glaciers (totaling 181,342 km$^2$). In fact, 63% of the Greenland ice sheet is drained by those 42 glaciers.

We use a multi-year average of the surface velocity in 250 m resolution from the MEaSUREs (Making Earth System Data Records for Use in Research Environments)[84] Greenland Ice Velocity data to assess the velocities of the glacier catchments. We calculate the average velocity of each catchment below 2000 m above sea level. We find that land-terminating glaciers have an average velocity of 28 m yr$^{-1}$ and marine-terminating glaciers have an average velocity of 107 m yr$^{-1}$.

## Reporting summary

Further information on research design is available in the Nature Portfolio Reporting Summary linked to this article.

## Data availability

The data generated in this study are provided in the Supplementary Information and Figshare data repository https://doi.org/10.6084/m9.figshare.23254361[42] Source data are provided with this paper.

## Code availability

Code or new software was not developed for this study. Mathematical calculations were undertaken by using the open-source programming Python.

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

## Acknowledgements

C.S.A. acknowledges funding from the Independent Research Fund Denmark (0217-00244B), the Research Council of Norway (NFR 324520) and the VILLUM Foundation (YIP 10100). F.S. acknowledges funding from NSF (grants 2020547 and 2127241). N.B.K. acknowledges funding from VILLUM Foundation (40858) and from PROMICE, which is funded by the Geological Survey of Denmark and Greenland (GEUS) and the Danish Ministry of Climate, Energy and Utilities under the Danish Cooperation for Environment in the Arctic (DANCEA) and is conducted in collaboration with DTU Space (Technical University of Denmark) and Asiaq, Greenland.

## Author contributions

C.S.A. designed the study and performed the analysis in collaboration with N.B.K., N.D., A.A.B., and I.E.G. Calculations of SAR for the period 1950–2009 were undertaken by T.J.A., S.S., and E.F.E. N.B.K. calculated the sediment flux and erosion rates. C.S.A. led the fieldwork for retrieving most of the sediment cores and wrote the manuscript with contributions from N.B.K., F.S., S.S., T.J.A., E.F.E., A.A.B., N.D., L.M.D., F.V., and I.E.G.

## Competing interests

The authors declare no competing interests
