## [Peer Review File · Nature Communications]

Sediment discharge from Greenland's marine-terminating glaciers is linked with surface meltREVIEWER COMMENTS

Reviewer #1 (Remarks to the Author):

Dear Editor, dear Authors,

The manuscript by Andresen et al. deals with a very important question about the sediment discharge from the Greenland ice sheet to the surrounding ocean. The study follows the recent work by Overeem et al. (2017) focused on sediment supply from the land-terminating margins of Greenland, and focus on dominating system – marine tidewater glaciers. So, the topic is certainly important not only from geological, oceanographic, and glaciological perspectives but also from biological and societal. However, the answer to the question is largely affected by the difficulty of actually measuring sediment fluxes, difficult access to marine tidewater margins, and the complexity of the systems. These factors cause that although the authors provide some numbers on potential sediment flux from the Greenland ice sheet – it is really hard to assess their quality, as they are limited only to 30 sediment cores (including 8 new ones) from several fjords, which are located more than 20 km from the nearest ice margin, many factors were assumed unimportant or even missed and the final key hypothesis would require much stronger arguments. The presented results are also difficult to be assessed as the original data are partly not presented. Some of the problems I have stated below. Nevertheless, it is also so far the best and the only assessment of this type for the Greenland margin, however, in my opinion, it should be presented in a much more critical way. In my opinion, the amount of data, although significant and certainly enlarging our knowledge, is still too small to address the question of sediment discharge from the Greenland Ice Sheet.

Some of the suggestions, which are in my opinion important to be considered include:

- It is a bit confusing throughout the paper to guess what is the new contribution. The results are mainly in the supplementary material. In various parts of the paper, the reader may read about 30, 27 and 8 new sediment cores. It seems that the new data are only for 8 cores, while the remaining cores are reused already published data, which in case 19 cores are recalculated into mass accumulation rates. It is also stated that transects from 9 fjords were studied, however, in some cases only single cores were dated from them. Actually, only two fjords were studied along the transects of more than 5 cores per fjord, results from these fjords also revealed the biggest scatter in the results. Thus it should suggest that also in the remaining fjords, a larger variability in sediment accumulation rate may be expected.
- The key methods used are the ^{210}Pb and ^{137}Cs dating. However, they are presented in a very limited way. There is no information even on such basic facts as the method applied (alpha or gamma spectrometry?). How the excess ^{210}Pb was obtained? There is no discussion on the applied age models. The data are presented only for the new cores in the form of a figure in the supplementary material. The modern recommendations are to present the actually measured activities, so that other researchers may assess the models or reuse the data (e.g. Mustaphi et al. 2019 <https://www.sciencedirect.com/science/article/abs/pii/S1871101418301249>). Moreover, some of the presented excess ^{210}Pb profiles reveal variable downcore profiles, some researchers would consider them 'undatable'. Probably it is safer to assess from them only the minimum sediment accumulation rate (based on the downward penetration depth of ^{137}Cs and excess ^{210}Pb). There is also a lack of critical assessment of the reused excess ^{210}Pb profiles.
- The next problem with dating is related to the comparison of sediment accumulation rates for variable time spans. Although, the Authors claim that it 20th century averaged sediment accumulation rate, in most of the cases the cores do not cover the complete 20th century. Usually, it is the last 60-80 years. It should be corrected. I suggest focusing on the post ~1955, which is often correlated with the penetration depth of ^{137}Cs and would correspond to the used data on surface melt of the glaciers. The Authors also present in the supplementary file the X-radiography of the cores, some of them seem to be laminated. Is it possible that the laminations could be annual, would it be worth using them for the accumulation rate assessment? Assessment of dry bulk density using the water content and assumed density is also a potential source of uncertainty.
- The Authors miss in the discussion of several potentially important factors. E.g., they do not take into

account the input of surface melt to the fjord through supraglacial and englacial channels (in particular the latter may provide meltwater without sediments), they do not take into account the additional processes responsible for the sediment transport and redeposition (currents, mass wasting processes, turbidite currents, subaqueous deltaic deposition in front of ice cliff etc.), they do not consider in their equations and interpretation the input from fjord side glaciers, they miss the key zone of proximal 20 km from the glacier fronts (remembering that the sedimentation is expected to decrease with distance in an exponential way). Most of the cores are also taken from the center line – the slope deposits are often redeposited and the center line of the fjords often represents a sediment focusing area. Taking into account that almost half of the studied cores are from a single fjord (Sermilik), which also shows the biggest variability in the accumulation rate, it is really difficult to be convinced by the final estimations of the total sediment discharge from Greenland spanning variability of fjords and climatic regimes.

- The Authors attempted to eliminate iceberg melting as a sediment source by using the sand fraction. It is important to underline that the icebergs contain all the fractions (same as glacial till), also the fine-grained. It is also important to notice that the coarser fraction is mainly concentrated in the basal ice layer, which is at the margin of the iceberg, so it is likely melted out mainly closer to the source. So, although I agree with the general assessment of the bigger importance of meltwater discharge for sedimentation, the reasoning needs improvement.

- In the abstract, introduction, and discussion many general overstating expressions are used ('immense importance', 'comprehensive marine sediment core dataset' etc). Taking into account many limitations of the study, there are not adequate.

- Throughout the study the authors claim to study sediment flux or (as in the title) sediment discharge. Actually, they present sediment accumulation rates in the part of several fjords (missing the part with presumably the highest accumulation rate next to the margins of the glaciers). Of course, one may argue that the presented values are representing the sediment flux to the seafloor. However, taking into account the underlined importance of the potential influence of sediment discharge on the biological productivity in the water column, it is probably not the case. So, I suggest clearly stating if the Authors mean sediment flux (horizontal) or sediment accumulation rates.

- The Authors underline the importance of the sediment discharge for biota. However, the elements discussed in the introduction (e.g. Fe) are in bioavailable forms mainly in dissolved load, which is not studied or discussed at all.

- It would be useful to attempt to make attempt to calculate sediment yield from the fjords catchments (although it would be with large uncertainty) and then to try to correlate it with surface melt. This could help to assess if there are significant differences in erosion rates, which could be related to other factors than surface melt only (type of bedrock etc.)

- I do not recommend creating a new abbreviation 'MTG' – it is not used so frequently and it is easier to read it in full.

- I do not fully agree with the number of statements, for instance: 'there is an empirical relationship between marine-terminating outlet glacier surface meltwater runoff and discharge of the sediment from beneath the glacier to a specific down-fjord distance'(the meltwater runoff and sediment discharge are not actually measured), 'enabling calculation of annual sediment flux at any point along a glacial fjord' (even the presented new data prove to represent many exceptions)

- Fig. 1: It is a bit misleading to start the scale from 20 km from the ice front instead of 0 km. The scale on the right panel is poorly visible, the same with the red arrow. Why ER11-26 is excluded? If it is not on the main track of icebergs, then the Author's assumptions on the negligible influence of icebergs on sedimentation and on the steady exponential decrease of sedimentation rate from suspension with distance should be well fitting to the core site.

- Fig. 2: In panel b is presented well-known rule about the decrease of sedimentation rate with distance. It is good to see it but it is not a particularly new discovery.

Concluding, I consider this work to be an important step toward the quantification of sediment discharge from the Greenland Ice Sheet, however, in my opinion, it is premature to assess it for all the Greenlandic fjords.

Reviewer #2 (Remarks to the Author):

This was a fascinating paper which analysed records of sediment mass accumulation (MAR) in different Greenland fjords, with a more in-depth study at Helheim glacier. An impressive number of cores were analysed and this is the first study of its kind to attempt to relate MAR to in fjord processes and glacier discharge. As such it is highly novel and certainly warrants publication if several points can be addressed. There were a number of minor weaknesses in the manuscript, largely related to how the work and study sites are described (these can be easily rectified). In addition, the goal of relating subglacial discharge to MAR I felt was unclear and warranted a deeper analysis and clearer reporting of data. The future projection of sediment export from the ice sheet I also felt was speculative as written at present.

1. Capitalise Greenland Ice Sheet throughout
2. Line 66 – should this be better referred to as “proglacial rivers”? “discharge of surface melt at depth” is also confusing and might be better rephrased only as “Subglacial discharge” (as indicated in brackets in the manuscript). “Glacier faces” might be better phrased as terminal ice cliffs or glacial margins if that is the meaning.
3. Line 83 – replace three/fourths with three quarters or 75%
4. MTG and Marine Terminating Glaciers both used throughout the manuscript. Would suggest abbreviating the first time with full description of the term, and use MTGs thereafter for consistency.
5. Line 103 – change to ambient “fjord” waters
6. Line 108-111. Partial repetition of lines 98-100. Merge for simplicity/conciseness.
7. Line 113 “The sediment deposited in close proximity to the glacier is primarily composed of fine-grained laminated clays and silts that are deposited at relatively high rates”. Quote rate range and make this more quantitative. What are these high rates relative to? 100x outer fjord?
8. Line 12 “While the accumulation rate of sand is roughly constant down-fjord.” It would be better to say there is no significant relationship between the MAR for sand and distance down fjord.
9. Line 121. Although it is obvious to me from the map where the ice mélange is, it would be useful to state where this is present in this fjord in the text. Since the first cores are c. 30 km down fjord from the glacier terminus, I guess that a wider region is also included in the statement around missing higher fluxes from the inner fjord, in which case it would be better to be more specific on this point.
10. Line 122 – to what degree do the authors believe that the icebergs also do not supply fine grained material (i.e. silt and clay) in addition to sand? And therefore, is the iceberg sediment MAR significantly underestimated?
11. Figure 1 – be useful if the scale bar in panel b. could be made a little clearer as its very hard to read.
12. Line 127 – “Our data clearly indicate that the primary sediment transporter to the fjord is the surface melt discharge at the glacier’s grounding zone, which is then carried by subglacial discharge plumes through fjord circulation, transporting the sediment far downstream”. This reads a little confusingly to me and is a common thread throughout the paper. It is subglacial discharge/meltwater that carries the sediment, which is routed into the fjord at MTGs initially via buoyant meltwater plumes, and then by fjord circulation. Strictly speaking surface meltwater on Greenland has almost no sediment, so using this term might be confusing to a reader. Can this be rephrased to “subglacial discharge”?
13. Line 149 – remove comma after Upernavik
14. Line 155 – what is meant by “the most productive glacier”?
15. Line 157 – how were these retreat events assessed in terms of their timing? What is “a collapse event”?
16. Line 174 – “However, the magnitude of the 20th Century MAR varies between fjords”. It would be better to say “However, the magnitude of the 20th Century MAR at a given distance from the glacier varies between fjords” or that there is considerable scatter in the relationship shown in Figure 2b.
17. Line 182 – delete “in the following” to simplify the sentence.
18. Line 186 – change p values to p value

19. Line 201 – “The empirical relationship outlined in Equation (1) is a function solely of glacier melt processes”. This equation includes distance from the glacier which would encompass fjord-related factors in the MAR, e.g. sedimentation, entrainment by currents etc. Therefore, this does not seem a correct statement.

20. Line 236 - “In addition to providing an observationally constrained estimate of sediment flux from MTGs, we also demonstrate that a significant portion of the sediment flux from the ice sheet is confined to a broad 80 km zone, coinciding with an area of high primary production¹⁷. Ref. 17 is used here but this only refers to Godthåbsfjord – is it the case that this zone in all the fjords the authors look at has high PP? Suggest re-phrase if this is not known or not substantiated by evidence.

21. Line 247 – “Our findings underscore the substantial influence of surface melt on the discharge of sediment from underneath the MTGs, indicating that an increase in surface runoff will result in a commensurate rise in sediment flux into the waters surrounding Greenland”. It is difficult to determine whether this is supported by the data as it is presented at the moment. Figure 2 shows that there is a weak relationship between MAR and distance downfjord, highlighting the role of fjord sedimentation processes but the importance of other factors. This relationship improves when the MAR is normalized by the glacial runoff 1950-2009, suggesting that large glaciers with high runoff may export more sediment to fjords and vice versa (and thus, the glacial runoff is an important driver). However, could it also be that the circulation dynamics in fjords with high subglacial meltwater discharge promote greater transport of SPM to more distal fjord areas (i.e. the zones of deposition shift, causing scatter in the relationship in Figure 2b)? There is also considerable variability in the MAR at given points for single fjords systems (e.g. Helheim) which warrant further mention in the manuscript as they imply that the sampling scheme used has an impact on the dataset, and thus conclusions drawn. It would also be useful to add the equation of the line in Figure 2c, R² and p value.

It would be helpful to know what the 1950-2009 runoff estimates were for each of the glaciers/fjords for comparison. I can see there is a table at the end of the manuscript but the formatting obscures the column labels. It would be great to include readable column titles, fjord names, the glacial runoff estimate for each fjord (1950-2009), the MAR and normalized MAR. It would also be very nice to see the MAR data from the Little Ice Age compared to the 20th Century (which the authors note that they analyse but which I could not find in the manuscript).

22. The prediction of future sediment export from MTG fjords in Table 1 seems a little speculative based on the data at present. I can see why it is a useful exercise/thought experiment, but it would be useful if the authors can fully justify their assumption that a. the sediment flux will increase linearly with glacial runoff in individual fjords and b. why the claimed relationship between glacial runoff and MAR across their multi-fjord dataset (i.e. in Figure 2c) would hold for individual fjords/glaciers in different melt years.

Reviewer #3 (Remarks to the Author):

Summary:

The Greenland ice sheet is postulated to be an extremely efficient erosional agent, supplying the coastal regions of Greenland with up to ~10% of the global river sediment supply from ~1% of total riverine runoff. Most of the previous estimates of sediment flux are derived from satellite observations of land terminating glaciers and their outlet rivers, coupled with a handful of in-situ “ground-truthed” samples. To date there are no observations of sediment export (and therefore erosional rates) of marine terminating glaciers due to the logistical challenges of sampling outflow, couple with the complex delivery and movement of sediment-rich meltwater plumes as they mix with seawater. This is problematic as marine terminating glaciers around Greenland supply much of the meltwater delivered to the coastal zone. The manuscript by Andresen et al. provides the first estimates of sediment export from marine terminating glaciers based on observational data from coastal sediment cores. Using

upscaling techniques, the authors hypothesize that marine terminating glaciers supply over 1 Gt of sediment to the coastal environment around Greenland every year, which exceeds previous estimates based on data from land terminating glaciers alone. Much of this sediment is trapped in the inner fjord region (<80 km from glacier front) due to rapid removal as meltwaters mix with high salinity seawater.

General comments:

The manuscript is timely, interesting, and highly valuable. I enjoyed reading the results and, even though they didn't really surprise me, it's nice to see some high-quality data to be able to make more accurate flux estimates of sediment export to the ocean in Greenland. The upscaling is a bit of a leap given the complexity of these systems, but without additional data (which hopefully this will provide the motivation to collect), I'm relatively happy with how it has been handled – I appreciate how difficult it is to collect this type of data in these remote locations. My comments/suggestions are therefore mostly specific to certain sections/sentences, and I feel the manuscript is already close to publication quality for Nature Communications.

My first general comment is that I think the spatial data is perhaps not as well utilized "up front" in the manuscript as it could be. The finding that much of the sediment is contained within fjords is not surprising, but it does have implications for marine elemental cycling that many would be interested in and should therefore probably be mentioned in the abstract. Conversely, as the authors find that some sediment makes it to the fjord mouth >80 km downstream of input, that's also an interesting secondary finding.

My second general comment is that this manuscript still feels a little like a transfer from Nat Geo (I'm assuming it is, but have not previously reviewed it). There's more room in Nat Comms to flesh out some of the ideas and arguments, and develop some of the methods, but I don't think this has been taken advantage of yet. The authors have an opportunity to include more figures and tables in the manuscript and I think should take advantage of this, and some of the methods are not currently adequately described (e.g., the measurements for Pb and Cs geochronology).

Specific comments/suggestions:

L22: Be more specific here – how does it influence biological systems? Only needs a little more text to give better context.

L27: I would be more specific than "numerous locations"

L29: Find a significant correlation between?

L32: "from Greenland's marine terminating glaciers..."

L39: "sediment archives"?

L58-59: This is dissolved and particulate silicon, so should be rephrased.

L62: I think it would be defining this as "total reactive phosphorus"

L84-85: "ice sheet melt and MTG sediment flux"

L199: "and were therefore..."

L174-180: I think this section needs some work to develop this hypothesis. There is no mention of glacier velocity, nor the geology of the underlying rock, nor the subglacial hydrology of these systems, all of which are likely to exert some kind of influence on the sediment flux (see Cowton et al., 2012, example below for example, which actually pushes back against the meltwater availability hypothesis). What I'm trying to say, is that I don't think it's as simple as argued here, and if it is, then the argument needs to be made clearer by discussing all the potential variables that could impact sediment mobilization and export.

L182-184: This paragraph should be incorporated into another – one sentence is too short. There also needs to be some explanation of how surface area of each of these catchments was chosen as this is non-trivial.

L207: How?

L220: I'm not sure it aligns well, as they're quantifying different pools. You could say that this is similar to a previous estimate for the whole of the Greenland ice sheet, like you say in the next sentence. You could remove this sentence and include that previous value in the next sentence instead.

L226: Do you have evidence/a reference for this. I'm not sure this is true – many land terminating glaciers around Greenland have very high erosive power (see Cowton et al., below, and the Overeem paper that is well referenced), and variable erosive power could also be a common trait of MTGs (but data is sparse). Some land terminating glaciers also have very large catchment areas so I don't think they can be generalized as they are in this sentence, particularly without any evidence given for this.

L235-241: This paragraph needs some development as it gets a little hand wavy, and confuses some of the complex fjord processes that are at play and lead to changes in productivity. You could keep this quite general by saying that sediment fluxes are likely to influence biological productivity in fjords, but the processes at play are still poorly understood due to complex confounding factors like nutrient induced upwelling along with increased sediment flux.

L243-245: Should be incorporated into another paragraph

L256: Recommend citing Tréguer et al. (2021) here as a more recent reference

Figure 2 and Table 1: What does KNS refer to? This looks like Nuup Kangerlua fjord?

Some suggestions for further reading/references:

Cowton, T., P. Nienow, I. Bartholomew, A. Sole, and D. Mair (2012) Rapid Erosion beneath the Greenland Ice Sheet. *Geology* 40, no. 4: 343–46. <https://doi.org/10.1130/G32687.1>.

Gunnarsen, Klara Cecilia, Lars Stoumann Jensen, Minik T. Rosing, and Christiana Dietzen (2023) Greenlandic Glacial Rock Flour Improves Crop Yield in Organic Agricultural Production, Nutrient Cycling in Agroecosystems 126, no. 1: 51–66. <https://doi.org/10.1007/s10705-023-10274-0>.

Tréguer, P. J., J. N. Sutton, M. Brzezinski, M. A. Charette, T. Devries, S. Dutkiewicz, C. Ehlert, et al. (2021) Reviews and Syntheses: The Biogeochemical Cycle of Silicon in the Modern Ocean. *Biogeosciences* 18, no. 4: 1269–89. <https://doi.org/10.5194/bg-18-1269-2021>.

Author responses to reviewer comments

**‘Sediment discharge from Greenland’s marine terminating glaciers is linked with surface melt’
by Andresen et al.
NCOMMS-23-25194-T**

We thank the three reviewers for all the very valuable comments. We have now carefully addressed each of these and here provide a detailed account of the changes. Please take note that line numbers reference the text in the manuscript *with* track changes.

REVIEWER COMMENTS

Reviewer #1 (Remarks to the Author):

Dear Editor, dear Authors,

The manuscript by Andresen et al. deals with a very important question about the sediment discharge from the Greenland ice sheet to the surrounding ocean. The study follows the recent work by Overeem et al. (2017) focused on sediment supply from the land-terminating margins of Greenland, and focus on dominating system – marine tidewater glaciers. So, the topic is certainly important not only from geological, oceanographic, and glaciological perspectives but also from biological and societal. However, the answer to the question is largely affected by the difficulty of actually measuring sediment fluxes, difficult access to marine tidewater margins, and the complexity of the systems. These factors cause that although the authors provide some numbers on potential sediment flux from the Greenland ice sheet – it is really hard to assess their quality, as they are limited only to 30 sediment cores (including 8 new ones) from several fjords, which are located more than 20 km from the nearest ice margin, many factors were assumed unimportant or even missed and the final key hypothesis would require much stronger arguments. The presented results are also difficult to be assessed as the original data are partly not presented. Some of the problems I have stated below. Nevertheless, it is also so far the best and the only assessment of this type for the Greenland margin, however, in my opinion, it should be presented in a much more critical way. In my opinion, the amount of data, although significant and certainly enlarging our knowledge, is still too small to address the question of sediment discharge from the Greenland Ice Sheet.

Thank you for recognizing the multidisciplinary importance of the topic theme and highlighting that our results are so far the best and the only assessment of this type for the Greenland margin to date. This was indeed our primary aim. A secondary aim is to motivate that future data collection in Greenland fjords include sediment coring for MAR assessments, so that the science community can work together to continue to reduce uncertainties. We think that we have a good case for motivating this – however, we also acknowledge the reviewer concern, that we need to be more critical. Therefore, we have added sections to discuss the complexities and precautions in terms of the data interpretation and discuss processes other than those accounted for in the empirical relationship we provide. In addition, we have strengthened the argumentation to justify quantifying the Greenland Ice Sheet sediment flux. Please see comments below.

1. Some of the suggestions, which are in my opinion important to be considered include:

- It is a bit confusing throughout the paper to guess what is the new contribution. The results are mainly in the supplementary material. In various parts of the paper, the reader may read about 30, 27 and 8 new sediment cores. It seems that the new data are only for 8 cores, while the remaining cores are reused already published data, which in case 19 cores are recalculated into mass accumulation rates.

All MAR data (27 cores) are new, except the data from Smith et al. 2002 (3 cores): The MAR is calculated based on recalculations (27 cores) of the sedimentation rates of the period after 1950 (as suggested by reviewer 1) of 19 previously published ^{210}Pb profiles and 8 newly constructed ^{210}Pb profiles. These new sedimentation rates are combined with unpublished water content (27 cores). Thus, our analysis would not have been possible to undertake by using published records alone – the data simply do not exist.

To clarify that the MAR calculations are new, but based in many cases on previously published ^{210}Pb profiles, we have revised the MAR method section (line 407-420):

*“We have calculated mass accumulation rates for the later 20th century (1950-2009) from 27 marine sediment cores from fjords by marine terminating calving glaciers and have supplemented the data with previously reported mass accumulation rates from three additional cores (Table S1, Fig. S3). The selection criterion for our analysis is that the cores are taken within 100 km from the current margin and that ^{210}Pb , and in some cases also ^{137}Cs , profiles were available for the cores. These profiles are a combination of previously published and new measurements (Table S1, Table S2 and Fig. S1). Next, we have applied the Constant Flux-Constant Sedimentation (CF:CS) model to the ^{210}Pb profiles in all cores to derive the average sedimentation rate (SAR), specifically for the period 1950-2009, since most cores were retrieved 2009-2014. We derived the average mass accumulation rates (MAR, by using water content (Rodriguez et al. 2020), as: Average MAR ($\text{kg m}^{-2} \text{yr}^{-1}$) = average SAR (m yr^{-1}) * dry bulk density (kg m^{-3}). Dry bulk density is calculated as $((100 - \text{average \% water})/100) * 2650 \text{ kg m}^{-3}$, where 2650 kg m^{-3} is the density of quartz. The sediment water content was measured concurrently with sampling for ^{210}Pb analysis.”*

2. It is also stated that transects from 9 fjords were studied, however, in some cases only single cores were dated from them. Actually, only two fjords were studied along the transects of more than 5 cores per fjord, results from these fjords also revealed the biggest scatter in the results. Thus it should suggest that also in the remaining fjords, a larger variability in sediment accumulation rate may be expected.

We have now changed the sentence in the abstract (line 31-33):

“We examine dated sediment cores from transects through nine glaciated fjords” into “We examine dated sediment cores from nine glaciated fjords”

We agree that it would have been good with more data points from each of the fjords, to establish the relationship based on individual fjord/glacier systems. However, in return, we report from a relatively high number of glacier/fjord systems (9), and find that by normalizing with surface run off, we find a statistically significant relationship once we combine all the data. This finding point to the importance of surface run off to quantify sediment flux and is the motivation for the manuscript.

3. The key methods used are the ^{210}Pb and ^{137}Cs dating. However, they are presented in a very limited way. There is no information even on such basic facts as the method applied (alpha or gamma spectrometry?). How the excess ^{210}Pb was obtained? There is no discussion on the applied age models. The data are presented only for the new cores in the form of a figure in the supplementary material. The modern recommendations are to present the actually measured activities, so that other researchers may assess the models or reuse the data (e.g. Mustaphi et al. 2019 <https://www.sciencedirect.com/science/article/abs/pii/S1871101418301249>). Moreover, some of the presented excess ^{210}Pb profiles reveal variable downcore profiles, some researchers would consider them ‘undatable’. Probably it is safer to assess from them only the minimum sediment accumulation rate (based on the downward penetration depth of ^{137}Cs and excess ^{210}Pb). There is also a lack of critical assessment of the reused excess ^{210}Pb profiles.

We have now expanded the methodology of measuring ^{210}Pb and ^{137}Cs activities. The radiometric data/activities are now provided in an additional file, to be uploaded as a supplemental dataset and go into the SEANOE - Sea Open Scientific Data Publication <https://www.seano.org/>.

For editing of the method section, please see response to reviewer 1's comment 1 (and line 407-420 in manuscript). In the supplementary information we have added:

"The naturally-occurring ^{210}Pb ($T_{1/2} = 22.3$ years) is used to estimate sediment accumulation rates in the Greenland glacial fjords 1,2. SAR is determined from profiles of ^{210}Pb in excess ($^{210}\text{Pb}_{\text{ex}}$) of that supported within sediment, by the decay of its radioactive parent (^{226}Ra), using the constant flux constant sediment (CF:CS) model. The compilation includes calculations of MAR on 19 cores for which profiles of $^{210}\text{Pb}_{\text{ex}}$ were already published (Table S1), and on 8 cores whose profiles of $^{210}\text{Pb}_{\text{ex}}$ are presented here (Fig. S1). To calculate SARs since the 1950s on the 27 cores, an iterative approach was used. A first SAR was calculated on the whole profile and an age model was established based on the sampling year of the cores. This dating then allowed to select the levels corresponding to sediment deposited since 1950 to recalculate average SAR for the period 1950-2009. A few of the cores only reach back to around 1980 and for those the average accumulation rate for the entire core have been calculated.

Three labs (university of Copenhagen and Bordeaux, and Washington University) were involved in the determination of $^{210}\text{Pb}_{\text{ex}}$ on the 8 cores measured for this study: ER11-22, ER11-23, ER11-26, Cr01, Cr05, ER14, ER15 and ER16, all from Sermilik Fjord. Laboratory methodology used for determination of $^{210}\text{Pb}_{\text{ex}}$ on the remaining 19 cores is provided in the publications listed in Table S1.

At the university of Copenhagen and Bordeaux, measurements on dry sediments were done using gamma spectrometers, that have permitted for determination of ^{210}Pb , ^{226}Ra and ^{137}Cs (for further details see Ref (3) and Ref (4)). At Washington University, ^{210}Pb activities were determined by alpha spectrometry after a radiochemistry step according the procedure described by Ref (1) and Ref (2). A mean supported ^{210}Pb activity was calculated as the average of ^{210}Pb determinations in deep sediment layers below the zone of ^{210}Pb exponential decline (between 67 and 203.5 cm for CR01 and between 48 and 188 cm for CR05). Note that age modeling was not performed on these cores due to concerns about calculating inventories in light of 10 cm core shortening of Cr01 after core retrieval.

Radionuclide activities are provided in TableS2 and uploaded at the Sea Open Scientific Data Publication (seano.org)."

References:

1. Jaeger, J.M., and C.A. Nittrouer, 1999. Marine record of surge-induced outburst floods from the Bering Glacier, Alaska. *Geology*, 27 (9), 847-850.
2. Jaeger, J.M., and C.A. Nittrouer, 1999. Sediment deposition in an Alaskan fjord: Controls on the formation and preservation of sedimentary structures in Icy Bay. *Journal of Sedimentary Research*, 69, 1011-1026.
3. Andersen, T.J., 2017. Some practical considerations regarding the application of ^{210}Pb and ^{137}Cs dating to estuarine sediments. *Applications of paleoenvironmental techniques in estuarine studies*, pp.121-140.
4. Schmidt S. & De Deckker P. 2015. Present-day sedimentation rates on the southern and southeastern Australian continental margins. *Australian Journal of Earth Sciences* 62, 143-150, doi: 10.1080/08120099.2015.1014846

4. The next problem with dating is related to the comparison of sediment accumulation rates for variable time spans. Although, the Authors claim that it 20th century averaged sediment accumulation rate, in most of the cases the cores do not cover the complete 20th century. Usually, it is the last 60-80 years. It should be corrected. I suggest focusing on the post ~1955, which is often correlated with the penetration depth of ^{137}Cs and would correspond to the used data on surface melt of the glaciers.

We agree, this is a very good point. We have now calculated average accumulation rates for the period 1950 to coring date. The dating is based on the CF:CS model with regressions on the part of the core spanning the period since 1950. A few of the cores only reach back to around 1980 and for those the average accumulation rate for the entire core have been calculated. The chronologies are confirmed by Cs-137 data when available as Cs-137 was not detected in samples dated to before 1950.

5. The Authors also present in the supplementary file the X-radiography of the cores, some of them seem to be laminated. Is it possible that the laminations could be annual, would it be worth using them for the accumulation rate assessment?

An interesting idea. We looked at it but think that the distinction between individual lamina is not quite clear.

6. Assessment of dry bulk density using the water content and assumed density is also a potential source of uncertainty.

Use of water content as a proxy for dry bulk density is common practice. This requires the assumption that the void spaces in the seabed sediment matrix are filled with water, which is typically true. See also Rodriguez et al., 2020. (Rodriguez, A. B., McKee, B. A., Miller, C. B., Bost, M. C., & Atencio, A. N. (2020). Coastal sedimentation across North America doubled in the 20th century despite river dams. *Nature Communications*, 11(1), 3249.)

7. The Authors miss in the discussion of several potentially important factors. E.g., they do not take into account the input of surface melt to the fjord through supraglacial and englacial channels (in particular the latter may provide meltwater without sediments), they do not take into account the additional processes responsible for the sediment transport and redeposition (currents, mass wasting processes, turbidite currents, subaqueous deltaic deposition in front of ice cliff etc.), they do not consider in their equations and interpretation the input from fjord side glaciers, they miss the key zone of proximal 20 km from the glacier fronts (remembering that the sedimentation is expected to decrease with distance in an exponential way). Most of the cores are also taken from the center line – the slope deposits are often redeposited and the center line of the fjords often represents a sediment focusing area. Taking into account that almost half of the studied cores are from a single fjord (Sermilik), which also shows the biggest variability in the accumulation rate, it is really difficult to be convinced by the final estimations of the total sediment discharge from Greenland spanning variability of fjords and climatic regimes.

We have now added a new section to address these additional processes mentioned by the reviewer: Regarding **englacial and supraglacial transport of sediment**, we have added the following section (line 238-244): *“ In the normalization, we assume that all surface runoff enters the subglacial system and is discharged subglacially at the glacier front, which is a common assumption in most studies aiming to quantify meltwater outflux (for example, Mankoff et al., 2020). The assumption is because most of the englacial transport of water in Greenland glaciers takes place in moulins (vertical pipes connecting the surface of the ice with the bed) or in crevasses (Nienow et al. 2017). Studies have shown that surface meltwater penetrates to the bed of the ice comparatively fast (hours to days, Yang et al., 2018).”*

Regarding **additional processes responsible for the sediment transport and redeposition**, we have built to the existing section, which originally stated:

“The empirical relationship outlined in Equation (1) is a function solely of glacier melt processes. However, the influence from subsequent processes that could modulate the MAR at a certain location in the fjord also plays a role in the resultant relationship, even though they are not parameterized. These processes may include local sediment focusing/dispersal in response to fjord narrowing/widening, respectively, or the strength of the hydrographic regime at the seabed. Most of the subglacial discharge and thereby sediment flux occurs during summer when the plume is more likely to dominate the fjords’ hydrographic circulation regime, however, remotely sourced oceanic currents may also influence the MAR”

Now it is stated (line 271-283):

“The empirical relationship outlined in Equation (2) implies that sediment flux at a certain location in a glacial fjord is modulated primarily by glacier melt processes. Other processes may also influence the MAR at a certain location in the fjord. These processes may include local sediment focusing/dispersal in response to fjord narrowing/widening, respectively, or the strength of the hydrographic regime at the seabed. Implicitly, we are assuming that most of the sediment flux, which is associated with the subglacial discharge, occurs during summer when the plume is more likely to dominate the fjords’ circulation⁶⁸, however, other drivers of circulation in fjords may also influence currents and transport of sediments⁶⁸. For example, the relatively high MAR of sediment cores at the bend in Sermilik fjord (Fig. 1a core ER14) may also indicate a decrease in the current strength, as a result of complex bathymetry, may also play a role in the deposition of sediments here.”

In addition, we have added the following section (line 284-289):

“Through the examination of sedimentary facies, our analysis was specifically directed toward assessing the accumulation of sediment originating from the meltwater plume and iceberg rafting (Methods). Thus, the sediment we have tracked has not undergone redeposition due to mass wasting events or turbidite flows, which are important processes for near-bed transfer of glacial sediment into the marine environment⁶⁹. It’s important to underscore that these are secondary processes, and by omitting sediment with turbidites, we can focus on assessing the primary accumulation of plume and iceberg sediment.”

Regarding side glaciers: These are actually not omitted from the analysis. We do consider their run off in the normalization procedure. But for simplicity, we have calculated the distance to the main glacier. However, we have not communicated this well enough. We have therefore changed the sentence.

From:

“ To account for differences in surface runoff between glaciers, the later 20th century MAR values are thus in the normalized (Fig. 2C, Methods) by dividing MAR for individual core sites with the average surface runoff over the period 1950–2009⁵² from the glaciers that contribute sediment-loaded meltwater (Table S1).”

To (Line 235-238):

*“To account for differences in surface runoff between glaciers, the later 20th century MAR values are thus normalized (Fig. 2C, Methods) by dividing MAR for individual core sites with the average surface runoff over the period 1950–2009⁵² from **all the glaciers (i.e. including side glaciers)** that contribute sediment-loaded meltwater to the fjord (Table S1).”*

8. The Authors attempted to eliminate iceberg melting as a sediment source by using the sand fraction. It is important to underline that the icebergs contain all the fractions (same as glacial till), also the fine-grained. We have now rephrased to underline that icebergs also contain clay and silt; we understand it may not have been clear. Thus, we have changed the sentence.

From: *“The sand grains are too heavy to remain suspended in the meltwater plume and were transported to this part of the fjord by icebergs”*

To (line 138-140):

“The sand grains are too heavy to remain suspended in the meltwater plume and were therefore transported to this part of the fjord, together with clay and silt, in icebergs”.

9. It is also important to notice that the coarser fraction is mainly concentrated in the basal ice layer, which is at the margin of the iceberg, so it is likely melted out mainly closer to the source. So, although I agree

with the general assessment of the bigger importance of meltwater discharge for sedimentation, the reasoning needs improvement.

Observations of grain size distribution in Greenland icebergs and from basal ice is nearly non-existent, but we were able to find two studies. These two studies allow some additional argumentation for the influence on sedimentation of meltwater discharge.

We have therefore added a section (line 140-145):

"It is worth noting that the quantity of clay and silt transported by icebergs to this location is expected to be smaller than that carried by the plume. This expectation is grounded in reports of 50-80% sand in basal ice debris by Russell Glacier (Baltrunas et al. 2009) and 50-70% sand in icebergs sampled off Scoresbysund glaciers (Soltau, 2020). The sand content in the diamicton at core site ER07 and ER11 is only 20% (Andresen et al. 2012) and therefore suggests clay and silt input from the plume. However, future sediment sampling from icebergs (Hasholt et al. 2022) is needed to robustly quantify the grain size distribution in iceberg sediments."

10. In the abstract, introduction, and discussion many general overstating expressions are used ('immense importance', 'comprehensive marine sediment core dataset' etc). Taking into account many limitations of the study, there are not adequate.

We have removed overstating expressions like *immense* and *comprehensive from the manuscript*. We have also removed the word *strongly* from the title "*Sediment discharge from Greenland's marine-terminating glaciers is strongly linked with surface melt*"

11. Throughout the study the authors claim to study sediment flux or (as in the title) sediment discharge. Actually, they present sediment accumulation rates in the part of several fjords (missing the part with presumably the highest accumulation rate next to the margins of the glaciers). Of course, one may argue that the presented values are representing the sediment flux to the seafloor. However, taking into account the underlined importance of the potential influence of sediment discharge on the biological productivity in the water column, it is probably not the case. So, I suggest clearly stating if the Authors mean sediment flux (horizontal) or sediment accumulation rates.

We have revised the following sentence to state that we use the MAR to infer sediment:

From:

"Our data analysis allows us to quantify the average sediment flux and spatial pattern of sedimentation in Greenland fjords over most of the 20th Century"

To (line 102-103):

"We use the mass accumulation rate (MAR) to infer the average sediment flux and its spatial pattern of sedimentation in these fjords since 1950".

We have also changed the caption in Fig. 2

From: "*Sediment discharge from Greenland's MTGs is linked with surface melt*"

To (line 202-204):

"Sediment flux from Greenland's marine-terminating glaciers, as inferred from the mass accumulation rate within fjords adjacent to these glaciers, exhibits a clear connection with surface melt processes".

Throughout the manuscript we now consistently use the term sediment flux instead of sediment discharge.

12. The Authors underline the importance of the sediment discharge for biota. However, the elements

discussed in the introduction (e.g. Fe) are in bioavailable forms mainly in dissolved load, which is not studied or discussed at all.

A number of studies (referenced in the manuscript: Cape et al. 2019, Bhatia et al. 2013, Hawkings et al. 2014) show a link between glacial meltwater discharge from calving glaciers and e.g. iron in both dissolved and particulate form. Non-conservative behaviour was observed by the 79Fjord Glacier (Krisch et al. 2021) emphasizing the need for more knowledge. We have now highlighted in the study that more research is needed to clarify the factors controlling bioaccessibility, by adding to the sentence (Line 393-396): “*Similarly, increased sediment discharge from the Greenland Ice Sheet warrants an improved understanding of the factors that control Si and Fe bioaccessibility, **including in dissolved form**, as well as transport to and within the North Atlantic Ocean in terms of conserving marine resources far offshore Greenland.*”

13. It would be useful to attempt to make attempt to calculate sediment yield from the fjords catchments (although it would be with large uncertainty) and then to try to correlate it with surface melt. This could help to assess if there are significant differences in erosion rates, which could be related to other factors than surface melt only (type of bedrock etc.)

We agree that it would be useful (and highly interesting) to calculate the sediment yield from the individual glacier/fjord systems and then correlate the sediment yield with surface melt. However, this is unfortunately not possible with the available data. As the reviewer noted earlier, there are only two fjords with transects of more than five sediment cores (Helheim and Upernavik) and that is not a sufficient number of data points to construct the correlation with surface melt on an individual fjord/glacier basis. As noted above, we hope that our findings will motivate future data collection in Greenland fjords to also incorporate sediment coring so this relationship can be constructed in individual fjords. Our finding of an exponential relationship between core distance and MARnormalized in the *combined* data set will hopefully motivate future core collection.

But we thank the reviewer for their suggestion to include erosion rates. We have now used the empirical equation 1 to calculate average basin-wide erosion rates for each of the glaciers listed in Table 1 and included this in Table 1. We find our estimates are within the range of previously reported erosion rates for Greenland (Cowton et al. 2012, based on Hasholt, 1996 and Graham et al. 2023) and this strengthens the confidence in our approach. We have added the following to the text:

To the main text (line 309-322):

“ We use the relationship described by Equation (1) to calculate the total annual sediment flux from the nine glaciers in our study for the period 1950-2009. We do this by calculating the annual amount of sediment deposited per square meter of seabed along the fjord, and (assuming that this deposition rate is representative across the fjord), we estimate the total mass of sediment deposited (see Equation (3), Methods). This gives a total sediment flux from the nine glaciers of 0.210 ± 0.137 Gt yr⁻¹ (Table 1). We then leverage our estimated sediment yield to calculate the erosion rates of each glacier catchment (Table 1, Methods Table S1), giving estimates ranging from 0.04 mm yr⁻¹ (Upernavik catchments) to 0.4 mm yr⁻¹ (Sermeq Kujalleq catchment). The estimates represent catchment-wide averages and are likely to mask substantial spatial variability (cf. Cowton et al., 2012). Studies of the Greenland Ice Sheet have reported erosion rates of 1.0 ± 0.5 mm yr⁻¹ (average ice sheet erosion rate, Cowton et al., 2012), 0.08 and 0.17 mm yr⁻¹ (southwest and west Greenland land-terminating glaciers, respectively Hasholt, 1996, Cowton et al. 2012), and 0.26 mm yr⁻¹ (Sermeq Kujalleq Graham et al., 2023 (in review)). Thus, our results, based on sediment cores, are in good agreement with previous studies.

To the Methods section (line 484-486):

“ Estimation of erosion rates:

We convert our sediment flux values (Table 1) from Gt yr⁻¹ to m³ yr⁻¹ using a density of 2650 kg m⁻³. The average basin-wide erosion rate is estimated by dividing the sediment yield by the area of the glacier

catchment (Koppes et al., 2015). We use defined glacier catchments (Mouginot and Rignot, 2019) and in the cases where multiple glaciers discharge into the same fjord, we sum the glacier catchment areas. See Table S1 for the area of each glacier catchment.”

14. I do not recommend creating a new abbreviation ‘MTG’ – it is not used so frequently and it is easier to read it in full.

*We have now spelled out *marine-terminating glacier* and do not use the MTG abbreviation.*

15. I do not fully agree with the number of statements, for instance: ‘there is an empirical relationship between marine-terminating outlet glacier surface meltwater runoff and discharge of the sediment from beneath the glacier to a specific down-fjord distance’ (the meltwater runoff and sediment discharge are not actually measured), ‘enabling calculation of annual sediment flux at any point along a glacial fjord’ (even the presented new data prove to represent many exceptions)

We agree that we have not measured the actual sediment discharged at the base of the glacier. Instead, we derive the sediment flux using the MAR observation data from the fjords and argue that we can use MAR (from assessment of the sedimentary facies and depositional pattern) to infer sediment flux. Similarly, we agree that meltwater run off is not measured, but derived using temperature observations and a model that estimates surface run off (The Programme for Monitoring of the Greenland Ice Sheet (PROMICE) and Mankoff et al. 2020). Given the statement we present is not based on a theoretical model or hypothesis, but rather is based on a statistical relationship between variables that we derive from observations, we find it justifiable to refer to this relationship as empirical. Having said this, we also acknowledge the need, pointed out by the reviewer, for discussing the data more critically and describe in more depth the potential processes involved (under the ice sheet and in the fjords), including what is inferred. We have now done so throughout the manuscript, and kindly refer to the responses we give to comments addressing these themes.

16. Fig. 1: It is a bit misleading to start the scale from 20 km from the ice front instead of 0 km. The scale on the right panel is poorly visible, the same with the red arrow.

We agree with these three notions and have changed Fig. 1 accordingly, also including the conceptual figure.

17. Why ER11-26 is excluded? If it is not on the main track of icebergs, then the Author’s assumptions on the negligible influence of icebergs on sedimentation and on the steady exponential decrease of sedimentation rate from suspension with distance should be well fitting to the core site.

*We had initially removed ER11-26 due to its position slightly off from the *iceberg* route, but failed to underline it is also off the *melt water* route. However, we set up a selection criterion that all MAR values from cores within 100 km from a glacier margin goes into the analysis. Therefore, we agree that ER11-26 should not be excluded – and have now included it. We have changed the caption for Fig. 1.*

From:

“The red line indicates the main track of icebergs flowing out of the fjord⁴⁴, note that core ER11-26 is positioned off this track explaining the low MAR values.”

To (line 173):

“The red line indicates the main track of icebergs and meltwater flowing out of the fjord⁴⁴, note that core ER11-26 is positioned off this track explaining the low MAR values.”

18. Fig. 2: In panel b is presented well-known rule about the decrease of sedimentation rate with distance. It is good to see it but it is not a particularly new discovery.

Concluding, I consider this work to be an important step toward the quantification of sediment discharge from the Greenland Ice Sheet, however, in my opinion, it is premature to assess it for all the Greenlandic fjords.

This study is the first to-date to attempt to find a universal relationship between sediment accumulation rate and the subglacial discharge for marine-terminating glaciers and we expect that it can prompt further investigation and more data collection. To add further argumentation to why it is justifiable to upscale to all of Greenland's marine-terminating glaciers, we have added a section (line 324-331):

"We consider that Equation (1) captures the governing processes for sediment deposition from marine-terminating glaciers in general, and thus the proposed relationship between sediment deposition, distance to the glacier front, and surface runoff should apply to other marine-terminating glaciers that share similarities with the nine glaciers in this study. We posit that this is the case for all marine-terminating glaciers in Greenland, as they are typically characterized by high ice flow velocities and large catchments, and thus the available surface runoff will dictate the sediment flux from the glaciers. We can therefore apply Equation (1) to upscale our estimate for the entire section of the Greenland Ice Sheet that discharges through marine-terminating outlets."

As was also noted in the previous version of our manuscript, our upscaled estimate (from only marine-terminating glaciers) aligns overall well with the estimate from Overeem et al. 2017 (based on observations from land-terminating glaciers upscaled to the entire ice sheet). The fact that the two studies utilize fundamentally different types of observation and still provide estimates that are overall within each other's range suggest both methodologies provide useful estimates. But we do also suggest Overeem et al. 2017 may have underestimated the contribution from marine-terminating glaciers, and note to the reviewers that the authors themselves suggests this underestimation (and recommend observations from marine-terminating glaciers).

We have now added a section to argument why marine-terminating glaciers would produce more sediment, and why Overeem et al. 2018 may have underestimated it (line 337-349):

"In fact, the previous study (Overeem et al. 2017), which relied on satellite observations of sediment plumes, may not have fully accounted for sediment transport processes that occur deep in the fjord—e.g., sediment fallout within the water column before the plume reaches the surface where it can be detected by satellites. This study also found only a weak linear correlation between sediment concentration and melt water discharge. It is worth considering that while sediment transport (by surface runoff) controls sediment flux from marine-terminating glaciers, sediment availability rather than sediment transport may be the limiting factor for land-terminating glaciers. On average, the catchments of marine-terminating glaciers are 20% larger than those of land-terminating glaciers, and average velocities are twice as high (Methods). This implies an overall greater sediment production by erosion and a larger catchment area for marine-terminating glaciers. We acknowledge that the erosion rate values presented here (Table 1) and previously reported (Cowton et al. 2012, Hasholt et al. 1996, Graham et al. 2023) represent large local variability and do not exclude that individual land-terminating glaciers may have high erosion rates"

Reviewer #2 (Remarks to the Author):

This was a fascinating paper which analysed records of sediment mass accumulation (MAR) in different Greenland fjords, with a more in-depth study at Helheim glacier. An impressive number of cores were analysed and this is the first study of its kind to attempt to relate MAR to in fjord processes and glacier discharge. As such it is highly novel and certainly warrants publication if several points can be addressed. There were a number of minor weaknesses in the manuscript, largely related to how the work and study sites are described (these can be easily rectified). In addition, the goal of relating subglacial discharge to MAR I felt was unclear and warranted a deeper analysis and clearer reporting of data. The future projection of sediment export from the ice sheet I also felt was speculative as written at present.

Thank you for the positive assessment of the study, including the acknowledgement of the large amount of

data that has gone into it and the novelty of our approach. We have now elaborated on the discussion of relating subglacial discharge to MAR in the context of processes taking place within and under the ice sheet, as well as processes taking place in the fjord. We have also elaborated on the discussion of projected sediment export from the Ice Sheet. Our revisions are discussed under the individual reviewer points.

1. Capitalise Greenland Ice Sheet throughout

Done.

2. Line 66 – should this be better referred to as “proglacial rivers”? “discharge of surface melt at depth” is also confusing and might be better rephrased only as “Subglacial discharge” (as indicated in brackets in the manuscript). “Glacier faces” might be better phrased as terminal ice cliffs or glacial margins if that is the meaning.

Good points. The sentence “*Sediment from the Greenland ice sheet is delivered to the ocean via terrestrial rivers, discharge of surface melt at depth (subglacial discharge) and submarine melting of icebergs and glacier faces*” has now been changed to (line 74-76): “*Sediment from the Greenland Ice Sheet is delivered to the ocean via proglacial rivers, subglacial discharge at depth and submarine melting of icebergs and glacial margins*”.

3. Line 83 – replace three/fourths with three quarters or 75%

We now write three-quarters (line 75).

4. MTG and Marine Terminating Glaciers both used throughout the manuscript. Would suggest abbreviating the first time with full description of the term, and use MTGs thereafter for consistency.

We now spell it out every time to avoid using abbreviations in the manuscript.

5. Line 103 – change to ambient “fjord” waters

Done (line 119).

6. Line 108-111. Partial repetition of lines 98-100. Merge for simplicity/conciseness.

We do understand the idea of merging here. However, we find that the first mentioning (line 113-114) serves just as an introduction to be followed by a characterization of the setting of Helheim Glacier. The second mentioning (line 124-126) details what was done. We find this introductory section important to keep the reader on track with the coming results.

7. Line 113 “The sediment deposited in close proximity to the glacier is primarily composed of fine-grained laminated clays and silts that are deposited at relatively high rates”. Quote rate range and make this more quantitative. What are these high rates relative to? 100x outer fjord?

We have added the values for the cores in the manuscript (line 133 and 135).

8. Line 120 “While the accumulation rate of sand is roughly constant down-fjord.” It would be better to say there is no significant relationship between the MAR for sand and distance down fjord.

Agreed – we changed the sentence (line 147) to “*While there is no significant relationship between accumulation rate of sand and core site distance to glacier margin*”

9. Line 121. Although it is obvious to me from the map where the ice mélange is, it would be useful to state where this is present in this fjord in the text. Since the first cores are c. 30 km down fjord from the glacier terminus, I guess that a wider region is also included in the statement around missing higher fluxes from the inner fjord, in which case it would be better to be more specific on this point.

Agreed – we now added (Line 149) that the mélange extends 10-20 km down-fjord of the glacier margin.

10. Line 122 – to what degree do the authors believe that the icebergs also do not supply fine grained material (i.e. silt and clay) in addition to sand? And therefore, is the iceberg sediment MAR significantly underestimated?

Reviewer 1 also pointed this part out (reviewer comment 8 and 9). We do believe that the icebergs also supply fine grained material and point this out more explicitly now. This is our response to reviewer 1:

- We have changed the sentence: *“The sand grains are too heavy to remain suspended in the meltwater plume and were transported to this part of the fjord by icebergs”* to (line 138-140): *“The sand grains are too heavy to remain suspended in the meltwater plume and were therefore transported to this part of the fjord, together with clay and silt, in icebergs”*.
- We have added an argument, that the iceberg sediment MAR is not significantly underestimated: (line 140-145): *“It is worth noting that the quantity of clay and silt transported by icebergs to this location is expected to be smaller than that carried by the plume. This expectation is grounded in reports of 50-80% sand in basal ice debris by Russel Glacier (Baltrunas et al. 2009) and 50-70% sand in icebergs sampled off Scoresbysund glaciers (Soltau, 2020). The sand content in the diamicton at core site ER07 and ER11 is only 20% (Andresen et al. 2012) and therefore suggests clay and silt input from the plume. However, future sediment sampling from icebergs (Hasholt et al. 2022) is needed to robustly quantify the grain size distribution in iceberg sediment.”*

11. Figure 1 – be useful if the scale bar in panel b. could be made a little clearer as its very hard to read. Thank you for noticing, we have made it larger now.

12. Line 127 – “Our data clearly indicate that the primary sediment transporter to the fjord is the surface melt discharge at the glacier’s grounding zone, which is then carried by subglacial discharge plumes through fjord circulation, transporting the sediment far downstream”. This reads a little confusingly to me and is a common thread throughout the paper. It is subglacial discharge/meltwater that carries the sediment, which is routed into the fjord at MTGs initially via buoyant meltwater plumes, and then by fjord circulation. Strictly speaking surface meltwater on Greenland has almost no sediment, so using this term might be confusing to a reader. Can this be rephrased to “subglacial discharge”?

Yes. The sentence mentioned above has now been changed to (line 156-159): *“Our data indicate that the primary sediment transporter to the fjord is the subglacial discharge at the glaciers grounding zone, which is then routed into the fjord via buoyant meltwater plumes and through fjord circulation, transporting the sediment over considerable distances downstream.”*

13. Line 149 – remove comma after Upernavik
Thank you for noticing – comma removed.

14. Line 155 – what is meant by “the most productive glacier”?
We have clarified and now write (line 190): *“the most melt water and iceberg producing glacier”*.

15. Line 157 – how were these retreat events assessed in terms of their timing? What is “a collapse event”?
Since we, in accordance with suggestion of reviewer 1, now report MAR since 1950 (and thus not the entire 20th century), this line has been removed.

16. Line 174 – “However, the magnitude of the 20th Century MAR varies between fjords”. It would be better to say “However, the magnitude of the 20th Century MAR at a given distance from the glacier varies between fjords” or that there is considerable scatter in the relationship shown in Figure 2b.
Agreed, we have changed the manuscript according to the above suggested sentence (line 217).

17. Line 182 – delete “in the following” to simplify the sentence.

Done.

18. Line 186 – change p values to p value

Done.

19. Line 201 – “The empirical relationship outlined in Equation (1) is a function solely of glacier melt processes”. This equation includes distance from the glacier which would encompass fjord-related factors in the MAR, e.g. sedimentation, entrainment by currents etc. Therefore, this does not seem a correct statement.

Yes, this sentence reads a little misleading. To clarify, we have now changed the sentence to (line 271-274):
“ The empirical relationship outlined in Equation (2) implies that sediment flux at a certain location in a glacial fjord is modulated primarily by glacier melt processes. Other processes may also influence the MAR at a certain location in the fjord.”

Following we stress the importance from fjord-related processes and have edited and added (line 275-289):
“These processes may include local sediment focusing/dispersal in response to fjord narrowing/widening, respectively, or the strength of the hydrographic regime at the seabed. Implicitly, we are assuming that most of the sediment flux, which is associated with the subglacial discharge, occurs during summer when the plume is more likely to dominate the fjords’ circulation (Jackson et al. 2014), however, other drivers of circulation in fjords may also influence currents and transport of sediment (Jackson et al. 2014). For example, the relatively high MAR of sediment cores at the bend in Sermilik fjord (Fig. 1a core ER14) may also indicate a decrease in the current strength, as a result of complex bathymetry, may also play a role in the deposition of sediments here.

Through the examination of sedimentary facies, our analysis was specifically directed toward assessing the accumulation of sediment originating from the meltwater plume and iceberg rafting (Methods). Thus, the sediment we have tracked has not undergone redeposition due to mass wasting events or turbidite flows, which are important processes for near-bed transfer of glacial sediment into the marine environment (Pope et al. 2019). It's important to underscore that these are secondary processes, and by omitting sediment with turbidites, we can focus on assessing the primary accumulation of plume and iceberg sediment.”

20. Line 236 - “In addition to providing an observationally constrained estimate of sediment flux from MTGs, we also demonstrate that a significant portion of the sediment flux from the ice sheet is confined to a broad 80 km zone, coinciding with an area of high primary production¹⁷. Ref. 17 is used here but this only refers to Godthåbsfjord – is it the case that this zone in all the fjords the authors look at has high PP? Suggest re-phrase if this is not known or not substantiated by evidence.

We have now reformulated the sentence to highlight that this is an observation from several fjords around Greenland and provide the references for this statement (line 299-302): *“In addition to providing an observationally constrained estimate of sediment flux from marine-terminating glaciers, we also demonstrate that a significant portion of the sediment flux from the ice sheet is confined to a 80-100 km zone down-fjord, coinciding with an area of increased nutrient upwelling which has been linked to high primary productivity in glacial fjords around Greenland (Meire et al. 2017, Oliver et al. 2020 and 2023, Kanna et al. 2018”.*

21. Line 247 – “Our findings underscore the substantial influence of surface melt on the discharge of sediment from underneath the MTGs, indicating that an increase in surface runoff will result in a commensurate rise in sediment flux into the waters surrounding Greenland”. It is difficult to determine whether this is supported by the data as it is presented at the moment. Figure 2 shows that there is a weak relationship between MAR and distance downfjord, highlighting the role of fjord sedimentation processes but the importance of other factors. This relationship improves when the MAR is normalized by the glacial runoff 1950-2009, suggesting that large glaciers with high runoff may export more sediment to fjords and

vice versa (and thus, the glacial runoff is an important driver). However, could it also be that the circulation dynamics in fjords with high subglacial meltwater discharge promote greater transport of SPM to more distal fjord areas (i.e. the zones of deposition shift, causing scatter in the relationship in Figure 2b)? There is also considerable variability in the MAR at given points for single fjords systems (e.g. Helheim) which warrant further mention in the manuscript as they imply that the sampling scheme used has an impact on the dataset, and thus conclusions drawn. It would also be useful to add the equation of the line in Figure 2c, R2 and p value.

Agreed, we have now expanded on the section discussing fjord-related processes that could modulate MAR (line 275-289 – see also response to reviewers 2's comment no. 19).

We have also added the equation of the line in Figure 2C and have added the R2 values. We now report the 95% confidence intervals generated by MATLAB and Python (stated in the method section). In the previous version we reported confidence intervals generated by Grapher.

It would be helpful to know what the 1950-2009 runoff estimates were for each of the glaciers/fjords for comparison. I can see there is a table at the end of the manuscript but the formatting obscures the column labels. It would be great to include readable column titles, fjord names, the glacial runoff estimate for each fjord (1950-2009), the MAR and normalized MAR. It would also be very nice to see the MAR data from the Little Ice Age compared to the 20th Century (which the authors note that they analyse but which I could not find in the manuscript).

All these datasets, including the Little Ice Age data, are in the Supplementary TableS1, but were obscured in the formatting to pdf, which we unfortunately had not noticed. Hopefully, the pdf is readable in this version, otherwise we refer the reviewer to the uploaded excel file (also to be made publicly available in the Pangea repository. We also note that the Little Ice Age data are shown in Fig2b.

22. The prediction of future sediment export from MTG fjords in Table 1 seems a little speculative based on the data at present. I can see why it is a useful exercise/thought experiment, but it would be useful if the authors can fully justify their assumption that a. the sediment flux will increase linearly with glacial runoff in individual fjords and b. why the claimed relationship between glacial runoff and MAR across their multi-fjord dataset (i.e. in Figure 2c) would hold for individual fjords/glaciers in different melt years.

a. Studies suggest that for some glaciers erosion rates increase exponentially with precipitation (Cook et al. 2020). However, these findings are uncertain and debated therefore we use a linear scaling as a conservative estimate. Cook et al., 2020, The empirical basis for modelling glacial erosion rates, Nature Comm., 11:759. <https://doi.org/10.1038/s41467-020-14583-8>.

To address this, we have added (line 379-381): *“Previous studies have suggested an exponential relationship between erosion rates and precipitation implying that an increase in surface runoff will lead to a substantial increase in sediment flux (Cook et al. 2020). Here we take a conservative approach and assume that the sediment flux increases linearly with runoff volumes.”*

b. We argue, based on the empirical relationship, that glacial run off is the main modulator (although not the sole modulator) and that sediment availability underneath the ice sheets marine-terminating glaciers is not a limiting factor for the sediment flux into the fjord. If so, the sediment flux can be assessed from any marine-terminating glacier when the surface run off during a given year is known (can be assessed from PROMICE data).

To strengthen the argument, we have mostly added and slightly edited a section (line 218-233):

“We hypothesize that this difference is primarily due to the substantial variation in subglacially discharged surface runoff from the different glaciers over the later 20th Century. In other words, we posit that sediments are abundant under marine-terminating glaciers, due to their high flow velocities (allowing substantial erosion) and large catchments (implying a large amount of available sediments). Thus, the limiting factor for sediment flux into the fjords is likely the transport mechanism that moves the sediments

to the glacier front. Sediments are thought to be influenced by a variety of processes on their pathway from initial erosion of bedrock (Koppes et al. 2009, 2015, Overeem et al, 2017), through release from the subglacial system (Delaney et al. 2019, Beaud et al. 2018), and during their transport and accumulation within the fjord system. For example, the hydrological regime and porewater pressure underneath the ice in response to surface melt (Shepherd et al. 2009, Bartholomew et al. 2010, Pimentel et al. 2011), influence ice dynamics and basal sliding (Ultee et al. 2022), creating feedback to the ice velocity and erosion. Regardless of these complexities in subglacial sediment dynamics (Alley et al. 1997, Swift et al. 2005) the common driver/modulator is surface melt. Thus, it is reasonable to attribute MAR differences between fjords to differences in surface runoff to the fjords. This link between sediment flux and surface melt is also in line with the fact that MAR values are an order of magnitude lower during the colder Little Ice Age (1200-1900 CE Kjær et al. 2022), when subglacial runoff and calving were markedly lower than today (Reeh et al. 2004, Andresen et al. 2017, Wangner et al. 2018) (Fig. 2b)’’.

Reviewer #3 (Remarks to the Author):

Summary:

The Greenland ice sheet is postulated to be an extremely efficient erosional agent, supplying the coastal regions of Greenland with up to ~10% of the global river sediment supply from ~1% of total riverine runoff. Most of the previous estimates of sediment flux are derived from satellite observations of land terminating glaciers and their outlet rivers, coupled with a handful of in-situ “ground-truthed” samples. To date there are no observations of sediment export (and therefore erosional rates) of marine terminating glaciers due to the logistical challenges of sampling outflow, couple with the complex delivery and movement of sediment-rich meltwater plumes as they mix with seawater. This is problematic as marine terminating glaciers around Greenland supply much of the meltwater delivered to the coastal zone. The manuscript by Andresen et al. provides the first estimates of sediment export from marine terminating glaciers based on observational data from coastal sediment cores. Using upscaling techniques, the authors hypothesize that marine terminating glaciers supply over 1 Gt of sediment to the coastal environment around Greenland every year, which exceeds previous estimates based on data from land terminating glaciers alone. Much of this sediment is trapped in the inner fjord region (<80 km from glacier front) due to rapid removal as meltwaters mix with high salinity seawater.

General comments:

The manuscript is timely, interesting, and highly valuable. I enjoyed reading the results and, even though they didn’t really surprise me, it’s nice to see some high-quality data to be able to make more accurate flux estimates of sediment export to the ocean in Greenland. The upscaling is a bit of a leap given the complexity of these systems, but without additional data (which hopefully this will provide the motivation to collect), I’m relatively happy with how it has been handled – I appreciate how difficult it is to collect this type of data in these remote locations. My comments/suggestions are therefore mostly specific to certain sections/sentences, and I feel the manuscript is already close to publication quality for Nature Communications.

Thank you for providing this positive assessment of our study and for pointing out the high-quality of the data justifying our calculations of sediment flux from Greenland’s marine-terminating glaciers. We now describe the complexities of the processes taking place within and under the ice sheet, as well as in the fjord in relation to sediment transport. In similarity with reviewer 3 we hope to motivate further collection of such data, and advocate for this in the manuscript.

My first general comment is that I think the spatial data is perhaps not as well utilized “up front” in the manuscript as it could be. The finding that much of the sediment is contained within fjords is not surprising, but it does have implications for marine elemental cycling that many would be interested in and should

therefore probably be mentioned in the abstract. Conversely, as the authors find that some sediment makes it to the fjord mouth >80 km downstream of input, that's also an interesting secondary finding. We agree and have added the following sentence (Line 299-305): *"In addition to providing an observationally constrained estimate of sediment flux from marine-terminating glaciers, we also demonstrate that a significant portion of the sediment flux from the ice sheet is confined to a 80-100 km zone down-fjord, coinciding with an area of increased nutrient upwelling which has been linked to high primary productivity in glacial fjords around Greenland^{17,70-72}. Future ocean data sampling campaigns in Greenland fjords should ideally incorporate sediment coring for the assessment of additional MARs (especially from sites closer to the glaciers), and thereby enable the scientific community to continually enhance the reliability of these data."*

This is also highlighted in the abstract now (line 38): *"...vast quantity sediment is retained within a down-fjord zone spanning 80-100 kilometers."*

My second general comment is that this manuscript still feels a little like a transfer from Nat Geo (I'm assuming it is, but have not previously reviewed it). There's more room in Nat Comms to flesh out some of the ideas and arguments, and develop some of the methods, but I don't think this has been taken advantage of yet. The authors have an opportunity to include more figures and tables in the manuscript and I think should take advantage of this, and some of the methods are not currently adequately described (e.g., the measurements for Pb and Cs geochronology).

Generally, we have substantiated the discussion and added several sections to describe in more detail the subglacial and fjord processes. Please also see our responses to reviewers 1 and 2. In addition, we have now calculated the erosion rates, which is a very useful parameter for studies of ice sheet erosion and sediment production.

The notion about the geochronology is also pointed out by reviewer 1, and we now describe the geochronology in more detail (see also response to reviewers 1's comment no. 3).

Specific comments/suggestions:

L22: Be more specific here – how does it influence biological systems? Only needs a little more text to give better context.

We have now changed the abstract introductory sentence *"Sediment discharged from the Greenland Ice Sheet influences marine biological systems around Greenland."* to (line 24-27): *"Sediment discharged from the Greenland Ice Sheet delivers nutrients to marine biological ecosystems around Greenland, shapes seafloor habitats, and creates sedimentary archives of landscape change. Accurate estimations of sediment flux and dispersal into the ocean are key to interpreting these archives and their impacts on ecosystems."*

L27: I would be more specific than "numerous locations"

We now specify here (in the abstract) that it is 30 locations.

L29: Find a significant correlation between?

Yes, we now write (line 34): *"We find a statistically significant empirical correlation between mass accumulation rates, normalized by surface runoff, and distance down-fjord..."*

L32: "from Greenland's marine terminating glaciers..."

We have removed the apostrophe (line 37).

L39: "sediment archives"?

We changed 'mud' to 'sediment' (line 46).

L58-59: This is dissolved and particulate silicon, so should be rephrased.

This has now been corrected (line 66-68): “ *However, recent research has highlighted the potential for significant export of Si from the Greenland Ice Sheet, with studies showing that glacial meltwater can contain up to 0.20 Tmol year⁻¹ of dissolved Si, which corresponds to around 50% of the **dissolved Si** transported by Arctic rivers (etc.)*”.

L62: I think it would be defining this as “total reactive phosphorus”
We have changed according to this suggestion (line 71).

L84-85: “ice sheet melt and MTG sediment flux”
YES – we have changed according (line 97).

L119: “and were therefore...”
We have changed accordingly (line 140).

L174-180: I think this section needs some work to develop this hypothesis. There is no mention of glacier velocity, nor the geology of the underlying rock, nor the subglacial hydrology of these systems, all of which are likely to exert some kind of influence on the sediment flux (see Cowton et al., 2012, example below for example, which actually pushes back against the meltwater availability hypothesis). What I’m trying to say, is that I don’t think it’s as simple as argued here, and if it is, then the argument needs to be made clearer by discussing all the potential variables that could impact sediment mobilization and export.

We agree this could be substantiated. We have now changed the sentence: “*However, the magnitude of the 20th Century MAR varies between fjords. We hypothesize that this difference is primarily due to the substantial variation in subglacially discharged surface runoff from the different glaciers over the 20th Century. In other words, we posit that sediments are abundant under MTGs, due to their high flow velocities and large catchments*”

To (line 218-233):

“*We hypothesize that this difference is primarily due to the substantial variation in subglacially discharged surface runoff from the different glaciers over the later 20th Century. In other words, we posit that sediments are abundant under marine-terminating glaciers, due to their high flow velocities (allowing substantial erosion) and large catchments (implying a large amount of available sediments). Thus, the limiting factor for sediment flux into the fjords is likely the transport mechanism that moves the sediments to the glacier front. Sediments are thought to be influenced by a variety of processes on their pathway from initial erosion of bedrock (Koppes et al. 2009, 2015, Overeem et al, 2017), through release from the subglacial system (Delaney et al. 2019, Beaud et al. 2018), and during their transport and accumulation within the fjord system. For example, the hydrological regime and porewater pressure underneath the ice in response to surface melt (Shepherd et al. 2009, Bartholomew et al. 2010, Pimentel et al. 2011), influence ice dynamics and basal sliding (Ultee et al. 2022), creating feedback to the ice velocity and erosion. Regardless of these complexities in subglacial sediment dynamics (Alley et al. 1997, Swift et al. 2005) the common driver/modulator is surface melt. Thus, it is reasonable to attribute MAR differences between fjords to differences in surface runoff to the fjords. This link between sediment flux and surface melt is also in line with the fact that MAR values are an order of magnitude lower during the colder Little Ice Age (1200-1900 CE Kjær et al. 2022), when subglacial runoff and calving were markedly lower than today (Reeh et al. 2004, Andresen et al. 2017, Wangner et al. 2018) (Fig. 2b)*”.

We have also added this sentence (line 238-244): “ *In the normalization, we assume that all surface runoff enters the subglacial system and is discharged subglacially at the glacier front, which is a common assumption in most studies aiming to quantify meltwater outflux (for example, Mankoff et al., 2020). The assumption is because most of the englacial transport of water in Greenland glaciers takes place in moulins (vertical pipes connecting the surface of the ice with the bed) or in crevasses (Nienow et al. 2017). Studies*

have shown that surface meltwater penetrates to the bed of the ice comparatively fast (hours to days, Yang et al., 2018)."

L182-184: This paragraph should be incorporated into another – one sentence is too short.
Done (see line 236).

There also needs to be some explanation of how surface area of each of these catchments was chosen as this is non-trivial.

We have added the following sentence (line 239-242): *"In the normalization, we assume that all surface runoff produced over the glaciers' catchments (Karlsson et al. (2023), enters the subglacial system and is discharged subglacially at the glacier front, which is a common assumption in most studies aiming to quantify meltwater outflux (Mankoff et al, 2020)."*

L207: How?

The sentence here is: *"Most of the subglacial discharge and thereby sediment flux occurs during summer when the plume is more likely to dominate the fjords' hydrographic circulation regime, however, remotely sourced oceanic currents may also influence the regime and hence the MAR"*.

We have now expanded this section to reference different drivers of the circulation (line 271-289): *"The empirical relationship outlined in Equation (2) implies that sediment flux at a certain location in a glacial fjord is modulated primarily by glacier melt processes. Other processes may also influence the MAR at a certain location in the fjord. These processes may include local sediment focusing/dispersal in response to fjord narrowing/widening, respectively, or the strength of the hydrographic regime at the seabed. Implicitly, we are assuming that most of the sediment flux, which is associated with the subglacial discharge, occurs during summer when the plume is more likely to dominate the fjords' circulation (Jackson et al. 2014), however, other drivers of circulation in fjords may also influence currents and transport of sediment (Jackson et al. 2014). For example, the relatively high MAR in sediment cores at the bend in Sermilik fjord (Fig. 1a core ER14), may indicate a decrease in the current strength, as a result of complex bathymetry, that may also play a role in the deposition of sediments here.."*

L220: I'm not sure it aligns well, as they're quantifying different pools. You could say that this is similar to a previous estimate for the whole of the Greenland ice sheet, like you say in the next sentence. You could remove this sentence and include that previous value in the next sentence instead.

We now write (line 332-336): *"This estimate is within the range of a previous estimate of 1.154 +/- 0.636 Gt yr-1 (1999-2013) of sediment from the whole ice sheet²."* Following we discuss that these are different pools and the implications of our estimate (line 336-350).

L226: Do you have evidence/a reference for this (*"It is worth considering that sediment availability may act as a limiting factor for land-terminating glaciers due to their lower erosion potential"*). I'm not sure this is true – many land terminating glaciers around Greenland have very high erosive power (see Cowton et al., below, and the Overeem paper that is well referenced), and variable erosive power could also be a common trait of MTGs (but data is sparse). Some land terminating glaciers also have very large catchment areas so I don't think they can be generalized as they are in this sentence, particularly without any evidence given for this.

We have now added several sections discussing erosion potential and the differences between land and marine-terminating glaciers. Firstly, we have calculated average basin-wide erosion rates for each of the glaciers listed in Table 1 and included this in a table. We found our estimates are in good agreement with the few previous studies. Regarding the erosion potential, we have added the following to the text:

To the main text (line 309-322):

“ We use the relationship described by Equation (1) to calculate the total annual sediment flux from the nine glaciers in our study for the period 1950-2009. We do this by calculating the annual amount of sediment deposited per square meter of seabed along the fjord, and (assuming that this deposition rate is representative across the fjord), we estimate the total mass of sediment deposited (see Equation (3), Methods). , which yields an estimated This gives a total sediment flux from the nine glaciers of 0.210 ± 0.137 0.203 ± 0.125 Gt yr⁻¹ (Table 1 and Methods).

We then leverage our estimated sediment yield to calculate the erosion rates of each glacier catchment (Table 1, Methods Table S1), giving estimates ranging from 0.04 mm yr⁻¹ (Upernavik catchments) to 0.4 mm yr⁻¹ (Sermeq Kujalleq catchment). The estimates represent catchment-wide averages and are likely to mask substantial spatial variability (cf. Cowton et al., 2012). Studies of the Greenland Ice Sheet have reported erosion rates of 1.0 ± 0.5 mm yr⁻¹ (average ice sheet erosion rate, Cowton et al., 2012), 0.08 and 0.17 mm yr⁻¹ (southwest and west Greenland land-terminating glaciers, respectively Hasholt, 1996, Cowton et al. 2012), and 0.26 mm yr⁻¹ (Sermeq Kujalleq Graham et al., 2023 (in review)). Thus, our results, based on sediment cores, are in good agreement with previous studies.

To the Methods section (line 484-486):

“ Estimation of erosion rates:

We convert our sediment flux values (Table 1) from Gt yr⁻¹ to m³ yr⁻¹ using a density of 2650 kg m⁻³. The average basin-wide erosion rate is estimated by dividing the sediment yield by the area of the glacier catchment (Koppes et al., 2015). We use defined glacier catchments (Mouginot and Rignot, 2019) and in the cases where multiple glaciers discharge into the same fjord, we sum the glacier catchment areas. See Table S1 for the area of each glacier catchment.”

We also argue sediments are abundant under marine-terminating glaciers, but highlight that we cannot rule out that certain land-terminating glaciers have high erosive power. Specifically, when arguing for abundant sediments under marine-terminating glaciers, and the main modulator being surface run off, we write (line 218-233):

“In other words, we posit that sediments are abundant under marine-terminating glaciers, due to their high flow velocities (allowing substantial erosion) and large catchments (implying a large amount of available sediments). Thus, the limiting factor for sediment flux into the fjords is likely the transport mechanism that moves the sediments to the glacier front. Sediments are thought to be influenced by a variety of processes on their pathway from initial erosion of bedrock (Koppes et al. 2009, 2015, Overeem et al, 2017), through release from the subglacial system (Delaney et al. 2019, Beaud et al. 2018), and during their transport and accumulation within the fjord system. For example, the hydrological regime and porewater pressure underneath the ice in response to surface melt (Shepherd et al. 2009, Bartholomew et al. 2010, Pimentel et al. 2011), influence ice dynamics and basal sliding (Ultee et al. 2022), creating feedback to the ice velocity and erosion. Regardless of these complexities in subglacial sediment dynamics (Alley et al. 1997, Swift et al. 2005) the common driver/modulator is surface melt. Thus, it is reasonable to attribute MAR differences between fjords to differences in surface runoff to the fjords. This link between sediment flux and surface melt is also in line with the fact that MAR values are an order of magnitude lower during the colder Little Ice Age (1200-1900 CE Kjær et al. 2022), when subglacial runoff and calving were markedly lower than today (Reeh et al. 2004, Andresen et al. 2017, Wangner et al. 2018) (Fig. 2b)”.

Regarding the erosive power of land-terminating glacier, we have added the following section (line 340-349):

This study (Overeem et al. 2017) also found only a weak linear correlation between sediment concentration and melt water discharge. It is worth considering that while sediment transport (by surface runoff) controls sediment flux from marine-terminating glaciers, sediment availability rather than sediment transport may be the limiting factor for land-terminating glaciers. On average, the catchments of marine-terminating

glaciers are 20% larger than those of land-terminating glaciers, and average velocities are twice as high (Methods). This implies an overall greater sediment production by erosion and a larger catchment area for marine-terminating glaciers. We acknowledge that the erosion rate values presented here (Table 1) and previously reported (Cowton et al. 2012, Hasholt et al. 1996, Graham et al. 2023) represent large local variability and do not exclude that individual land-terminating glaciers may have high erosion rates”.

L235-241: This paragraph needs some development as it gets a little hand wavy, and confuses some of the complex fjord processes that are at play and lead to changes in productivity. You could keep this quite general by saying that sediment fluxes are likely to influence biological productivity in fjords, but the processes at play are still poorly understood due to complex confounding factors like nutrient induced upwelling along with increased sediment flux.

Yes. We follow this advice and have changed the sentence.

From:

“In addition to providing an observationally constrained estimate of sediment flux from MTGs, we also demonstrate that a significant portion of the sediment flux from the ice sheet is confined to a broad 80 km zone, coinciding with an area of high primary production¹⁷. With only a small fraction of sediment managing to reach the open ocean, being carried by currents or trapped in icebergs, fjords by MTGs are known to be crucial sites for the deposition of organic matter in sediments, which is then mineralized by microbial activity⁵³. As such, our sediment flux estimate can offer valuable insights for studies investigating the role of glacial fjords in the global carbon cycle^{12,24}.”

(line 299-302): *“In addition to providing an observationally constrained estimate of sediment flux from marine-terminating glaciers, we also demonstrate that a significant portion of the sediment flux from the ice sheet is confined to a 80-100 km zone down-fjord, coinciding with an area of increased nutrient upwelling which has been linked to high primary productivity in glacial fjords around Greenland (Meire et al. 2017, Oliver et al. 2020 and 2023, Kanna et al. 2018”.*

L243-245: Should be incorporated into another paragraph

Yes. The sentence has been split. The previous sentence was “The impact of future climate warming on glacier retreat and hydrological changes, and thereby subglacial sediment dynamics, is complex and multifaceted⁵⁴⁻⁵⁵ and involves changes in glacial erosion of bedrock^{28,56,2} sediment supply⁵⁷ and subglacial hydraulic gradients⁵⁸”. We now elaborate on the processes involved in subglacial sediment dynamics in a previous section (please see response to reviewer 3 above suggestion (L226) and address the consequences of increased surface run off in a later section (line 376-397): *“Our findings underscore the substantial influence of surface melt on the discharge of sediment from beneath the marine-terminating glaciers, indicating that an increase in surface runoff will result in a commensurate rise in sediment flux into the waters surrounding Greenland. In recent decades (2010-2020), the surface runoff has significantly increased with a decadal average of 230 km³ (Mankoff et al. 2020). Previous studies have suggested an exponential relationship between erosion rates and precipitation implying that an increase in surface runoff will lead to a substantial increase in sediment flux (Cook et al. 2020). Here we take a conservative approach and assume that the sediment flux increases linearly with runoff volumes. This implies an increase in sediment and nutrient transport from the ice sheet into the oceans in recent years, with an estimated increase of up to 45% and a combined total of 1.324 +/- 0.79 Gt yr⁻¹ currently being overturned from the marine-terminating sector (Table 1). This is consistent with observations indicating that the present-day sediment flux from Greenland is approximately 56% higher than during 1961–1990 (Overeem et al. 2017).”*

L256: Recommend citing Treguer et al. (2021) here as a more recent reference

Done, thank you for providing this, and also the other references.

Figure 2 and Table 1: What does KNS refer to? This looks like Nuup Kangerlua fjord?

KNS is Kangiata Nunaata Sermia, the most productive glacier in the Kangersuneq fjord (which is the inner fjord of Nuup Kangerlua). We have now spelled it out in the manuscript.

For references provided in the response letter, we kindly refer to the reference list in the revised manuscript.

REVIEWERS' COMMENTS

Reviewer #2 (Remarks to the Author):

I have read the response to the reviewers' comments and feel that they did a very good job in dealing with the various criticisms in the revised manuscript. I have no further comments and wish them the best of luck with the manuscript.

Reviewer #3 (Remarks to the Author):

I thank the authors for working hard to improve the manuscript. I am largely satisfied with the changes made and would therefore consider the manuscript ready for publication, bar the minors point below.

L66-68: I'm not sure the authors fully understood my point here. The flux given is not just dissolved Si, but a dissolved and pseudo-particulate Si flux. I'd recommend the authors change this to "0.20 Tmol yr⁻¹ of dissolved and dissolvable Si, which corresponds to 50% of the annual Si flux from Arctic rivers". This needs to be corrected before publication.

The authors did also not adequately provide details of how surface area of each of the glacier catchments was chosen (as per my comment "There also needs to be some explanation of how surface area of each of these catchments was chosen as this is non-trivial."). The current response does not provide a clear method to catchment area choice. I'm assume these are modeled catchment areas (preferential flow paths? surface topography? something else?)? Either way - this needs to be detailed as choosing catchment areas for ice sheets can be a tricky business. This is an especially pertinent point given the authors use the catchment areas to calculate erosion rates, and therefore needs to be adequately detailed before publication.

Author responses to reviewer comments

'Sediment discharge from Greenland's marine terminating glaciers is linked with surface melt' by
Andresen et al.
NCOMMS-23-25194-T

REVIEWERS' COMMENTS

Reviewer #2 (Remarks to the Author):

I have read the response to the reviewers' comments and feel that they did a very good job in dealing with the various criticisms in the revised manuscript. I have no further comments and wish them the best of luck with the manuscript.

Thank you.

Reviewer #3 (Remarks to the Author):

I thank the authors for working hard to improve the manuscript. I am largely satisfied with the changes made and would therefore consider the manuscript ready for publication, bar the minors point below.

Thank you.

L66-68: I'm not sure the authors fully understood my point here. The flux given is not just dissolved Si, but a dissolved and pseudo-particulate Si flux. I'd recommend the authors change this to "0.20 Tmol yr⁻¹ of dissolved and dissolvable Si, which corresponds to 50% of the annual Si flux from Arctic rivers". This needs to be corrected before publication.

Makes sense, we changed according to this suggestion

The authors did also not adequately provide details of how surface area of each of the glacier catchments was chosen (as per my comment "There also needs to be some explanation of how surface area of each of these catchments was chosen as this is non-trivial."). The current response does not provide a clear method to catchment area choice. I'm assume these are modeled catchment areas (preferential flow paths? surface topography? something else?)? Either way - this needs to be detailed as choosing catchment areas for ice sheets can be a tricky business. This is an especially pertinent point given the authors use the catchment areas to calculate erosion rates, and therefore needs to be adequately detailed before publication.

In the previous version we wrote: *We use defined glacier catchments⁸² to assess the glacier catchment size of marine-terminating and land-terminating glaciers.*

We have elaborated, so that it now states: *We use previously published glacier catchments⁸² to assess the glacier catchment size of marine-terminating and land-terminating glaciers. The catchments are defined using the direction of ice flow for fast-moving areas (>100m/yr) and the steepest surface slope for slower areas, where the surface slope is smoothed over 10 ice thicknesses (see Ref 82).*

Ref. 82: Mougnot, Jeremie; Rignot, Eric (2019). Glacier catchments/basins for the Greenland Ice Sheet [Dataset]. Dryad. <https://doi.org/10.7280/D1WT11>

final note from reviewer provided in author checklist: please clarify where data come from and what is new - sounds like material from 27 cores that didn't previously have accumulation rates was used, but not clear if ^{210}Pb and ^{137}Cs measurements were undertaken as part of this study (please clarify and have this information later in the manuscript – it should be together and succinct)

We now have reformulated the method section:

We have calculated mass accumulation rates for the later 20th century (1950-2009) from 27 marine sediment cores from fjords by marine terminating calving glaciers and have supplemented the data with previously reported mass accumulation rates from three additional cores (Table S1, Fig. S3). The selection criterion for our analysis is that the cores are taken within 100 km from the current margin. We have applied the Constant Flux-Constant Sedimentation (CF:CS) model to the ^{210}Pb profiles from the 27 cores to derive the average sedimentation rate (SAR) for the period 1950-2009, since most cores were retrieved 2009-2014. The sediment water content was measured concurrently with sampling for ^{210}Pb analysis, allowing us to calculate the average mass accumulation rates (MAR) from the 27 cores in the period 1950-2009 (Table S1) using Equation (3):

*Average MAR ($\text{kg m}^{-2} \text{ yr}^{-1}$) = average SAR (m yr^{-1}) * dry bulk density (kg m^{-3}). Dry bulk density is calculated as $((100 - \text{average \% water})/100) * 2650 \text{ kg m}^{-3}$, where 2650 kg m^{-3} is the density of quartz.*

The ^{210}Pb profiles that provide input to the SAR and MAR calculations includes previously reported measurements from 19 cores and new measurements from 8 cores (Table S2 and Fig. S1).